# Non-Supersymmetric AdS from String Theory

**Zihni Kaan Baykara**[1,2]**, Daniel Robbins**[3] **and Savdeep Sethi**[2]

[1] *Jefferson Physical Laboratory*
*Harvard University, Cambridge, MA 02138, USA*

[2] *Enrico Fermi Institute & Kadanoff Center for Theoretical Physics*
*University of Chicago, Chicago, IL 60637, USA*

[3] *Department of Physics*
*University at Albany, Albany, NY 12222 USA*

**Abstract**

We construct non-supersymmetric $AdS_3$ solutions of the $O(16) \times O(16)$ heterotic string. Most of the backgrounds have a classical worldsheet definition but are quantum string vacua in the sense that string loop corrections change the curvature of spacetime. At one-loop, the change in the cosmological constant is positive but never sufficient to uplift to de Sitter space. By computing the spectrum of spacetime scalars, we show that there are no tachyons below the BF bound. We also show that there is a solution with no spacetime NS-NS flux. This background has no classical string limit. Surprisingly, there appears to be parametric control over these constructions, which provide a framework for exploring quantum gravity and holography without supersymmetry.

# 1   Introduction

*Motivation*

The universe is accelerating [1]. A possible explanation for the dark energy driving this acceleration is vacuum energy. If this is the correct explanation then we must be able to construct long-lived de Sitter backgrounds in string theory. There has been much debate about past attempts to build de Sitter solutions; for a review, see [2]. Most such attempts are in the framework of superstring theory with supersymmetry restored at the string scale. Yet breaking supersymmetry in a controlled fashion is one of the fundamental problems with many of the attempted de Sitter constructions [3].[1] What if we have simply been looking in the wrong place? What if de Sitter constructions are more tractable in a framework which does not have any supersymmetry at the string scale?

We know remarkably little about string backgrounds without supersymmetry. This is for good reason. Unlike the case of superstrings, it is difficult to construct compactifications of non-supersymmetric strings which are even perturbatively stable. Scalar fields like the dilaton, which determines the string coupling, usually have a potential energy and must be stabilized in a regime of weak coupling using additional ingredients like fluxes. The resulting background often develops further pathologies like tachyonic instabilities.

One might have thought that holography would provide a definition of quantum gravity on AdS spacetime without supersymmetry. This would mean constructing conformal field theories with the right properties to define a weakly coupled gravitational theory. However, there are swampland conjectures suggesting that non-supersymmetric anti-de Sitter space is always unstable [6, 7], via non-perturbative instabilities of the type discussed in [8]. There is already some evidence that this conjecture is perhaps too strong. There are classical AdS$_4$ solutions of massive type IIA supergravity with no problematic tachyons [9], along with a continuous two parameter family of AdS$_4$ solutions in type IIB string theory whose

---

[1]There are studies of how resumming higher order corrections might give rise to string theory de Sitter solutions found in [4, 5].

perturbative stability is still being assessed [10]. There are also classical AdS$_3$ solutions of supergravity again with no problematic tachyons [11]. For at least some of these models, the status of non-perturbative instabilities is still unclear. Brane configurations in non-supersymmetric strings have also been examined [12, 13]. The branes appear to support non-supersymmetric conformal field theories at least in the large $N$ limit.

There is also a proposed construction of a large $N$ non-supersymmetric CFT obtained by deforming a supersymmetric CFT by a relevant double trace operator [14]. This is perhaps the most compelling currently proposed explicit CFT counter-example, though it is fair to say that it is not yet completely nailed down. For example, the existence of a CFT at finite $N$ is still unclear. There is simply much to be understood about how holography works in the absence of supersymmetry.

*Our construction*

These questions motivate us to revisit string theory without spacetime supersymmetry. In this work, we will provide a top down construction of non-supersymmetric anti-de Sitter space in the context of perturbative string theory. The construction is in the framework of the O(16) × O(16) heterotic string [15, 16], which is one of the ten-dimensional tachyon-free non-supersymmetric string theories. This string can be constructed as an orbifold of either the supersymmetric $E_8 \times E_8$ or Spin(32)/$\mathbb{Z}_2$ strings, or as an orbifold of a tachyonic string with a diagonal modular invariant [17].

The NS sector of the O(16) × O(16) heterotic string contains a metric $g$, a B-field $B_2$ and the string dilaton $\phi$. These are all the ingredients needed for this construction. The gauge-fields will play a minor role in our discussion. We consider the tree-level string geometry:[2]

$$\text{AdS}_3 \times S^3 \times \hat{S}^3 \times S^1 \,. \tag{1.1}$$

This background is characterized by three integers $(n_1, n_5, \hat{n}_5)$ which determine the amount of $H_3$-flux threading AdS$_3$, $S^3$ and $\hat{S}^3$, respectively. There is an exact conformal field theory description of this background at tree-level, which has been studied quite heavily in the context of the type II superstring; for a sampling of papers, see [18–21]. A discussion of AdS$_3$ in the heterotic superstring can be found in [22]. The exact tree-level description of (1.1) requires each flux quantum number $(n_1, n_5, \hat{n}_5)$ to be non-vanishing. The dilaton,

---

[2]To distinguish between the two spheres, we use a hat to denote the second sphere, $\hat{S}^3$, as well as the variables associated to it.

which determines the string coupling $g_s = e^\phi$, is also frozen at tree-level under the same conditions in terms of the flux integers:

$$g_s^4 \sim \frac{n_5^2 \hat{n}_5^2 (|n_5| + |\hat{n}_5|)}{n_1^2}. \tag{1.2}$$

For large $n_1$ with fixed $(n_5, \hat{n}_5)$, the string coupling becomes parametrically small. There are two classes of moduli visible in string perturbation theory. The first are universal moduli of the $O(16) \times O(16)$ string compactified on $S^1$. These moduli are the radius of the $S^1$ and the Wilson line moduli for the $O(16) \times O(16)$ gauge-fields. We will want these moduli to have no tadpole in our background; equivalently (1.1) should be an extremum of the spacetime potential energy, which depends on these scalar modes.

In principle, we do not even have to worry about stabilizing the Wilson line moduli when the $S^1$ is large because they are compact scalars, which cannot run away. Only the radius of the $S^1$ can lead to run away at the level of single trace operators visible in the world-sheet theory. However, all these perturbative moduli will be massed up by the string one-loop potential even when the radius of the $S^1$ is string scale. This one-loop potential for the $O(16) \times O(16)$ string was computed numerically for $\mathbb{R}^{10}$ in [15, 16] giving a string-frame potential

$$-\frac{1}{(2\pi\alpha')^5} \int d^{10}x \sqrt{-g} \Lambda \,, \tag{1.3}$$

where $\Lambda \sim 0.037$ and $1/(2\pi\alpha')$ is the string tension. Because the dilaton is frozen at tree-level in our construction, this can serve as an actual cosmological constant rather than a potential that might result in run away behavior for the dilaton. It was subsequently computed numerically for $\mathbb{R}^9 \times S^1$ in [23]. With modern technology, it is likely the potential can even be computed analytically for certain toroidal compactifications. More general compactifications of the $O(16) \times O(16)$ string were studied in [24, 25]. The $\mathbb{R}^9 \times S^1$ result should be a good approximation to the actual potential for the background (1.1) when the curvature scales of the AdS$_3$ and the spheres are low.

The second class of moduli involve deformations of the spheres themselves. There are marginal operators in the string worldsheet theory of the form $J\bar{J}'$, where $J$ and $\bar{J}'$ are holomorphic and anti-holomorphic currents, respectively, in the $SU(2) \times SU(2)$ WZW model describing $S^3 \times \hat{S}^3$. Turning on a small deformation of this type reduces the chiral algebra. It is natural to expect that a point of enhanced chiral symmetry is an extremum of the spacetime potential. A tadpole for a marginal operator of this type would then correspond to a spacetime coupling linear in a charged scalar field, which is ruled out by

gauge invariance [26]. This makes it quite plausible that (1.1) is also an extremum of the spacetime potential with respect to these non-universal marginal deformations. It would be nice to check this directly from the one-loop string vacuum energy computed for the full background (1.1).

There is one other potential subtlety we need to discuss. Since we are compactifying to three dimensions, the gauge-fields themselves can be dualized to scalar degrees of freedom, at least in principle. However these scalar modes should not be problematic: first, the modes are not visible in string perturbation theory. In addition, the scalars are compact and cannot lead to runaway. Lastly, it is reasonable to expect that any such modes will become massive in this non-supersymmetric theory, much like the Wilson line moduli.

There are also potential instabilities associated to multi-trace operators. These instabilities, which need not be large $N$ suppressed, have been seen by studying marginal multi-trace operators in three and four-dimensional gauge theories with broken supersymmetry [27, 28]. We examine the multi-particle states dual to these potentially troublesome operators in section 5.6. Fortunately most choices of quantization in $\text{AdS}_3$ do not appear to possess marginal multi-trace operators. Even the quantization choices that do give marginal operators look safe in these backgrounds because those multi-particle combinations are still charged under the spacetime gauge symmetry. There might also be non-perturbative instabilities in these backgrounds. This flavor of instability, should it is exist here, is likely to teach us something quite interesting about these non-supersymmetric AdS spaces.

*Intrinsically quantum string vacua*

We note that there has been very interesting work studying gravity solutions for the various currently known non-supersymmetric tachyon-free strings, by including the ten-dimensional disk or one-loop cosmological constant in the spacetime action [29–33]. Those solutions are typically of the form $\text{AdS} \times$ sphere. More often than not the resulting solution of the spacetime equations of motion has tachyonic instabilities, although it might be possible to remove those tachyons by a suitable projection in some cases [31].

The way the dilaton is stabilized in these constructions is by balancing the tree-level potential generated by the curvature of the sphere, threaded by $n$ units of flux, against the ten-dimensional flat space 1-loop or disk potential. This gives rise to an AdS solution with a string coupling that becomes weaker as $n$ becomes larger.

For example in the $\text{O}(16) \times \text{O}(16)$ heterotic string, there is an $\text{AdS}_7 \times S^3$ background with $g_s \sim n^{-1/2}$ and the radius $R$ of the sphere scaling like $R \sim n^{5/8}$ where $n$ is the amount

of $H_3$-flux threading the sphere; see, for example, [34]. Putting aside questions of bad tachyons below the BF bound [35], at large $n$ these backgrounds have weak curvature with a weak stabilized string coupling but have no corresponding tree-level string solution. We might call such backgrounds *intrinsically quantum* string vacua since they are found by balancing tree-level effects against a loop correction. We currently do not have technology for constructing string perturbation theory around such backgrounds, though that is a fascinating question.

This intrinsic case should be contrasted with the more conventional picture of a tree-level string solution, defining a worldsheet conformal field theory, which receives small loop corrections controlled by $g_s$. There is a procedure, in principle, for computing these corrections. We actually have both cases in our model. When the tree-level $g_s$ of (1.2) is small, we have a good weakly coupled tree-level background. However as we take $n_1 \to 0$, the tree-level $g_s$ would appear to blow up. At this point we meet a very exciting feature of the known tachyon-free non-supersymmetric string theories. They abhor strong coupling! The 1-loop potential simply prohibits the string coupling from becoming arbitrarily large. We might hope that the weak couplings actually seen in nature are in some way connected to this feature. In section 4, we find that for $n_5 = \hat{n}_5 = n$ the coupling scales as $g_s \sim |n|^{-1/2}$ for the intrinsically quantum case with $n_1 = 0$.

*Summary*

In this work, we use the $D = 10$ 1-loop potential energy of the $O(16) \times O(16)$ heterotic string combined with a spacetime effective field theory approach. However the exact 1-loop string Casimir energy can be studied in this background giving the 1-loop uplift as a function of the flux quantum numbers. That analysis merits a separate discussion and will appear elsewhere, along with an examination of potential non-perturbative stabilities for this family of backgrounds.

In this analysis we find that the cosmological constant is uplifted by the string 1-loop potential. One might have thought this uplift would be sufficient to guarantee a de Sitter solution since we can make the curvature of our initial AdS$_3$ arbitrarily small. The actual uplift term, however, is cleverer and becomes smaller and smaller as AdS$_3$ approaches flat space. We never uplift to de Sitter! This suggests that there might be a 1-string loop generalization of the tree-level no-go theorem found in [36]. It is also in accord with the two derivative gravity analysis of [37], which rules out de Sitter solutions when including the $D = 10$ 1-loop potential. We should stress that our result holds for a background with

a string scale circle and with the possibility of making the curvature of the spheres large, though that regime really requires a full string theory treatment.

The bulk of our analysis is studying whether there are tachyons below the BF bound, which would ruin perturbative stability. This is an involved analysis because there are tachyons precisely at the BF bound even in the supersymmetric theory with the background (1.1). The question for us is what happens to the masses of the scalar fields when we include the string 1-loop potential energy. In determining the fate of the scalars, we benefited greatly from the work done in [38] for the supersymmetric case. The end result is that we find no tachyons below the BF bound for our background, even including the intrinsically quantum case of $n_1 = 0$.

## 2 Effective theory of the $O(16) \times O(16)$ heterotic string

The tree-level action for the $O(16) \times O(16)$ heterotic string takes the form:

$$S_{D=10} = \frac{1}{2\kappa_{10}^2} \int d^{10}x \, e^{-2\phi} \sqrt{-g} \left( R + 4(\partial\phi)^2 - \frac{1}{12}|H_3|^2 + \dots \right) , \qquad (2.1)$$

$$\kappa_{10}^2 := \frac{1}{2}(2\pi)^7 \alpha'^4 e^{2\phi_0} . \qquad (2.2)$$

The omitted terms involve kinetic terms for the gauge-fields, which will not play a role in this discussion. We define $|H_3|^2 := H_{\mu\nu\rho}H^{\mu\nu\rho}$. We have factored out the dilaton expectation value $\phi_0$ which determines the string coupling $g_s = e^{\phi_0}$. The fluctuating dilaton $\phi$ has expectation value $\langle\phi\rangle = 0$.

The definition of $H_3$ involves the usual heterotic modification involving the Chern-Simons forms for the spin and gauge-connection, denoted $\omega$ and $\mathcal{A}$, respectively:

$$H_3 = dB_2 + \frac{\alpha'}{4} \left( \mathrm{CS}(\omega_+) - \mathrm{CS}(\mathcal{A}) \right), \qquad \omega_+ = \omega + \frac{1}{2}H . \qquad (2.3)$$

In our model, we will have at least the unbroken ten-dimensional $O(16) \times O(16)$ gauge symmetry. In fact, this gauge symmetry will be enhanced because the circle of $\mathrm{AdS}_3 \times S^3 \times \hat{S}^3 \times S^1$ will end up frozen at a special radius.

There are no 4-cycles in the background $\mathrm{AdS}_3 \times S^3 \times \hat{S}^3 \times S^1$ with non-vanishing $\mathrm{Tr}\,(R \wedge R)$ which would require a non-flat gauge bundle to ensure that $dH$ is trivial in cohomology. In Appendix A, we compute $\mathrm{CS}(\omega_+)$ for this background to leading order in $\alpha'$. It is non-vanishing, which means there are potential stringy corrections to the definition of $H_3$. However, there are also still residual choices for the flat gauge-bundle connection.

For example, we can choose the standard embedding on the spheres by identifying a sub-bundle of the $O(16) \times O(16)$ gauge connection with the spin connection. One can even do this in the AdS$_3$ space-time directions if we consider Euclidean AdS space. Another choice is to simply set the gauge connection to zero. This is consistent as long as the correct flux quantization condition is satisfied. For the moment let us simply ignore the Chern-Simons modification (2.3) to the definition of $H_3$. As long as any $H_3$-flux is supported on large cycles compared with the string scale, this is completely reasonable.

We want to reduce the tree-level $D = 10$ string-frame action (2.1) on $S^3 \times \hat{S}^3 \times S^1$ to a three-dimensional theory on $\mathcal{M}_3$. We take the following $D = 10$ metric,

$$ds^2 = ds^2_{\mathcal{M}_3} + e^{2\chi}d\Omega_3^2 + e^{2\hat{\chi}}d\hat{\Omega}_3^2 + e^{2\sigma}dx_{10}^2 , \tag{2.4}$$

where $d\Omega_3^2$ is the metric for a sphere with volume $2\pi^2 L^3$, $d\hat{\Omega}_3^2$ has volume $2\pi^2\hat{L}^3$ and the circle coordinate satisfies $x_{10} \sim x_{10} + 2\pi r$. The fields $(\chi, \hat{\chi}, \sigma)$ are scalar fields from the $D = 3$ perspective with expectation values

$$\langle \chi \rangle = \langle \hat{\chi} \rangle = \langle \sigma \rangle = 0 . \tag{2.5}$$

The first step is to reduce on $S^1$. At tree-level, this is an easy exercise. We obtain one additional vector-field from the metric $g$ and one from the $B$-field, along with scalar fields parametrizing the radius of the $S^1$ and the compact Wilson line moduli. The radius of $S^1$, given by the expectation value $r$, is a free parameter at tree-level. The resulting $D = 9$ effective action takes the form

$$S_{D=9} = \frac{1}{2\kappa_9^2} \int d^9x\, e^{-2\phi+\sigma}\sqrt{-g}\left(R + 4(\partial\phi)^2 - \frac{1}{12}|H_3|^2 + \dots\right) , \qquad \kappa_9^2 := \frac{\kappa_{10}^2}{2\pi r} , \tag{2.6}$$

where we continue to use $g$ to denote the now 9-dimensional metric in a slight abuse of notation.

String theory requires a quantized $H_3$-flux through the spheres which we fix by demanding that:

$$\frac{1}{4\pi^2\alpha'}\int_{S^3} H_3 = n_5 , \qquad \frac{1}{4\pi^2\alpha'}\int_{\hat{S}^3} H_3 = \hat{n}_5 , \qquad n_5, \hat{n}_5 \in \mathbb{Z} . \tag{2.7}$$

In terms of the volume form $\epsilon_{S^3}$ for the sphere of radius $L$, the internal flux takes the form

$$H_3^{\text{int}} = \frac{2\alpha' n_5}{L^3}\epsilon_{S^3} + \frac{2\alpha'\hat{n}_5}{\hat{L}^3}\epsilon_{\hat{S}^3} . \tag{2.8}$$

These internal fluxes together with the curvature of the spheres give one contribution to the effective potential. Reduction on the spheres gives a $D = 3$ theory with an Einstein-Hilbert term,

$$S_{D=3} = \frac{1}{2\kappa_3^2} \int d^3x \, \text{vol} \cdot \sqrt{-g_3} R + \dots, \qquad \text{vol} := e^{-2\phi} e^\sigma e^{3\chi} e^{3\hat{\chi}}, \tag{2.9}$$

$$\kappa_3^2 := \frac{\kappa_{10}^2}{(2\pi r)(2\pi^2 L^3)(2\pi^2 \hat{L}^3)}, \tag{2.10}$$

where $g_3$ now refers to remaining three-dimensional space-time metric.

To determine the potential energy, we will want to go to Einstein frame in $D = 3$ using the hatted metric $g_3 = \text{vol}^{-2}\hat{g}_3$ while holding fixed $\kappa_3^2$ to compute:

$$S_{D=3} = \dots - \frac{1}{2\kappa_3^2} \int d^3x \sqrt{-\hat{g}_3} V(\phi, \sigma, \chi, \hat{\chi}). \tag{2.11}$$

The internal flux and sphere curvature generated potential energy then takes the form,

$$V^{\text{int}}(\phi, \sigma, \chi, \hat{\chi}) = \text{vol}^{-2} \left\{ \frac{2(\alpha' n_5)^2}{L^6} e^{-6\chi} - \frac{6}{L^2} e^{-2\chi} + \frac{2(\alpha' \hat{n}_5)^2}{\hat{L}^6} e^{-6\hat{\chi}} - \frac{6}{\hat{L}^2} e^{-2\hat{\chi}} \right\}. \tag{2.12}$$

The $D = 3$ action also contains the $H_3$-flux kinetic term with action,

$$-\frac{1}{2\kappa_3^2} \int \text{vol}^4 \cdot \frac{1}{2} H_3 \wedge *H_3 + \frac{1}{2\kappa_3^2} \int d\left(\text{vol}^4 B_2 \wedge *H_3\right). \tag{2.13}$$

The second term is a total derivative needed to ensure that the variational problem is well-defined for this background with non-zero fundamental string charge [39, 40]. The net effect of this term is to change the sign of the contribution of the electric $H_3$-flux to the potential energy from what one might have expected naively. This is crucial to see the stabilization of the dilaton and match the physics of the full $D = 10$ solution.

Now we need to determine the contribution to the potential from the electric $H_3$-flux that threads the spacetime $\mathcal{M}_3$, which we denote $H_3^{\text{electric}}$. Quantization of this electric flux through $\mathcal{M}_3$ follows from quantization of the $D = 10$ dual field strength $H_7 = *\left(e^{-2\phi} H_3\right)$ through any 7-cycle $\Sigma_7$:

$$\frac{1}{(2\pi)^6 (\alpha')^3 g_s^2} \int_{\Sigma_7} H_7 = n_1, \qquad n_1 \in \mathbb{Z}. \tag{2.14}$$

In our case, $\Sigma_7 = S^3 \times \hat{S}^3 \times S^1$ and $H_7 = \frac{8\pi \alpha'^3 g_s^2}{r L^3 \hat{L}^3} n_1 \, \epsilon_{S^3} \cdot \epsilon_{\hat{S}^3} \cdot dx_{10}$. The corresponding electric $H_3^{\text{electric}}$ is given in terms of the volume form $\epsilon_3$ for $\mathcal{M}_3$ by

$$H_3^{\text{electric}} = \text{vol}^{-1} \frac{8\pi \alpha'^3 g_s^2}{r L^3 \hat{L}^3} n_1 \, \epsilon_3. \tag{2.15}$$

This is prior to going to Einstein frame in $D = 3$. On making that transformation and taking the total derivative term of (2.13) into account, we find the potential energy:

$$V(\phi, \sigma, \chi, \hat{\chi}) = \text{vol}^{-2} \left\{ \frac{2(\alpha' n_5)^2}{L^6} e^{-6\chi} - \frac{6}{L^2} e^{-2\chi} + \frac{2(\alpha' \hat{n}_5)^2}{\hat{L}^6} e^{-6\hat{\chi}} - \frac{6}{\hat{L}^2} e^{-2\hat{\chi}} \right\}$$
$$+ \text{vol}^{-4} \frac{(8\pi\alpha'^3 g_s^2)^2}{2r^2 L^6 \hat{L}^6} n_1^2 . \tag{2.16}$$

Extremizing this potential leads to three independent equations because $\phi$ and $\sigma$ only appear in the combination $\phi - \frac{1}{2}\sigma$, which determines the $D = 9$ string coupling. These equations have a unique solution which, once we impose (2.5), fixes the radii and nine-dimensional string coupling:

$$L = \sqrt{\alpha'|n_5|}, \qquad \hat{L} = \sqrt{\alpha'|\hat{n}_5|}, \qquad \frac{g_s^4}{r^2} = \frac{n_5^2 \hat{n}_5^2 (|n_5| + |\hat{n}_5|)}{16\pi^2 \alpha' n_1^2}. \tag{2.17}$$

For non-zero $n_5, \hat{n}_5$ and $n_1$, this is a minimum of the potential. There is always a flat direction parameterizing the radius of the $S^1$ and the Wilson line moduli. At this minimum, the value of the cosmological constant is

$$\Lambda = \frac{1}{2} V \Big|_{2\phi - \sigma = \chi = \hat{\chi} = 0} = -\left( \frac{1}{L^2} + \frac{1}{\hat{L}^2} \right) = -\frac{1}{\alpha'} \left( \frac{1}{|n_5|} + \frac{1}{|\hat{n}_5|} \right). \tag{2.18}$$

This determines the AdS length scale

$$L_{\text{AdS}}^2 := \left( \frac{1}{L^2} + \frac{1}{\hat{L}^2} \right)^{-1}. \tag{2.19}$$

For large values of $|n_5|$ and $|\hat{n}_5|$, the spheres of size $L$ and $\hat{L}$ are large compared to the string scale. This is the regime where we can trust this spacetime effective field theory approach. The scaling of the sphere size seen in (2.17) is in qualitative accord with our expectations from the associated SU(2) WZW models in the large radius gravity limit.

The cosmological constant can also be made as small as we like by choosing large $n_5$ and $\hat{n}_5$. It is important to note that we can do this while holding fixed the $D = 9$ string coupling determined by $\frac{g_s^2}{r}$. Conversely, the $D = 9$ string coupling given in (2.17) can be made as small as we want while holding fixed the cosmological constant (2.18) by making $|n_1|$ as large as we like.

Plugging in the parameter values from (2.17) into (2.16) gives a final $D = 3$ potential energy:

$$V(\phi, \sigma, \chi, \hat{\chi}) = \text{vol}^{-2} \frac{2}{\alpha'} \left( \frac{e^{-6\chi}}{|n_5|} - \frac{3e^{-2\chi}}{|n_5|} + \frac{e^{-6\hat{\chi}}}{|\hat{n}_5|} - \frac{3e^{-2\hat{\chi}}}{|\hat{n}_5|} \right) \tag{2.20}$$
$$+ \text{vol}^{-4} \frac{2}{\alpha'} \left( \frac{1}{|n_5|} + \frac{1}{|\hat{n}_5|} \right). \tag{2.21}$$

We have plotted the potential (2.20) versus $\phi$ in figure 1a and the potential versus $\chi$ in figure 1b for a large value of $n_5 = \hat{n}_5$. The dependence of the potential on $\sigma$ is similar in character to $\phi$ since both variables only appear in the combination $\phi - \frac{1}{2}\sigma$ in (2.20). The dependence on $\hat{\chi}$ is identical to $\chi$. This analysis is *identical* to what one would do in the supersymmetric case so it seems highly unlikely that any new instability would appear here. Indeed we have checked that the critical point is a local minimum.

## 3 One-loop construction

### 3.1 Vacuum solution

To complete the construction, we need to add the one loop potential. In principle, this can be computed in string theory precisely for the background (1.1). This computation is interesting in its own right and will appear elsewhere. For low curvatures, which correspond to large $n_5$ and $\hat{n}_5$, we can use the computation of the one-loop potential on $\mathbb{R}^9 \times S^1$ found in [23] along with effective field theory.

There are two key features of that potential: first, the radius is frozen at the self-dual value, so

$$r = \sqrt{\alpha'}. \tag{3.1}$$

The $\sigma$ field is now massed up. We have checked in Appendix B that this critical point is indeed a minimum as seen numerically in [23]. At this point, the gauge symmetry is enhanced to $SO(16) \times SO(16) \times SU(2)$. Fortunately all the Wilson line moduli are also massive at this point so we can ignore them.

In Appendix B, we describe the derivation of the $D = 3$ one-loop potential that results from compactification on $S^3 \times \hat{S}^3$. Here we will quote the result (B.36),

$$V^{1-\text{loop}} = e^{6\phi - 3\sigma - 6\chi - 6\hat{\chi}} \times 2\lambda \frac{g_s^2}{\alpha'}, \tag{3.2}$$

where $\lambda \approx 0.705$ is a dimensionless constant appearing in (B.30).

We can recompute the location of the shifted critical point. Minimizing the complete potential including the one-loop correction (3.2) will result in a modification in the values of the sphere sizes $L, \hat{L}$ and the string coupling $g_s$. We denote the one-loop corrected values with a loop subscript: $L_\circ, \hat{L}_\circ, g_{s,\circ}$.

By a redefinition of $\phi$ to $\phi - \frac{1}{2}\sigma$, we get rid of the $\sigma$ dependence in the potential. Next, we require that the total potential $V_\circ := V + V^{1-\text{loop}}$ maintains the minimum at $\phi = \chi =$

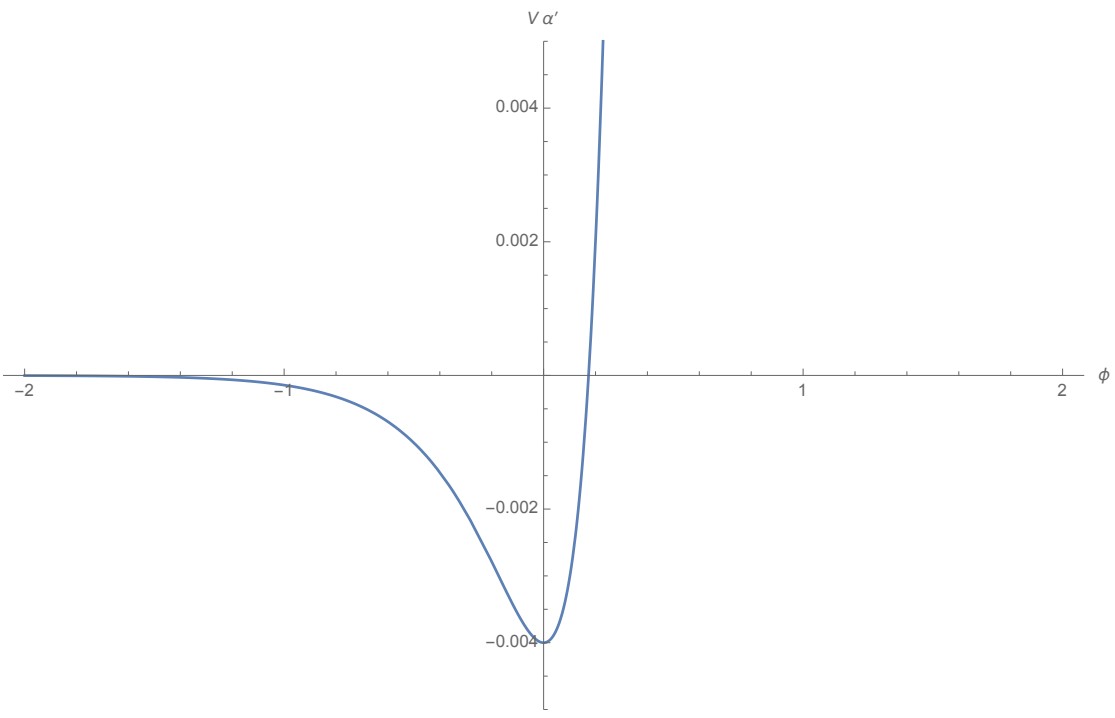

(a) The potential $V(\phi)$ with $\sigma = \chi = \hat{\chi} = 0$ and $n_5 = \hat{n}_5 = 10^3$.

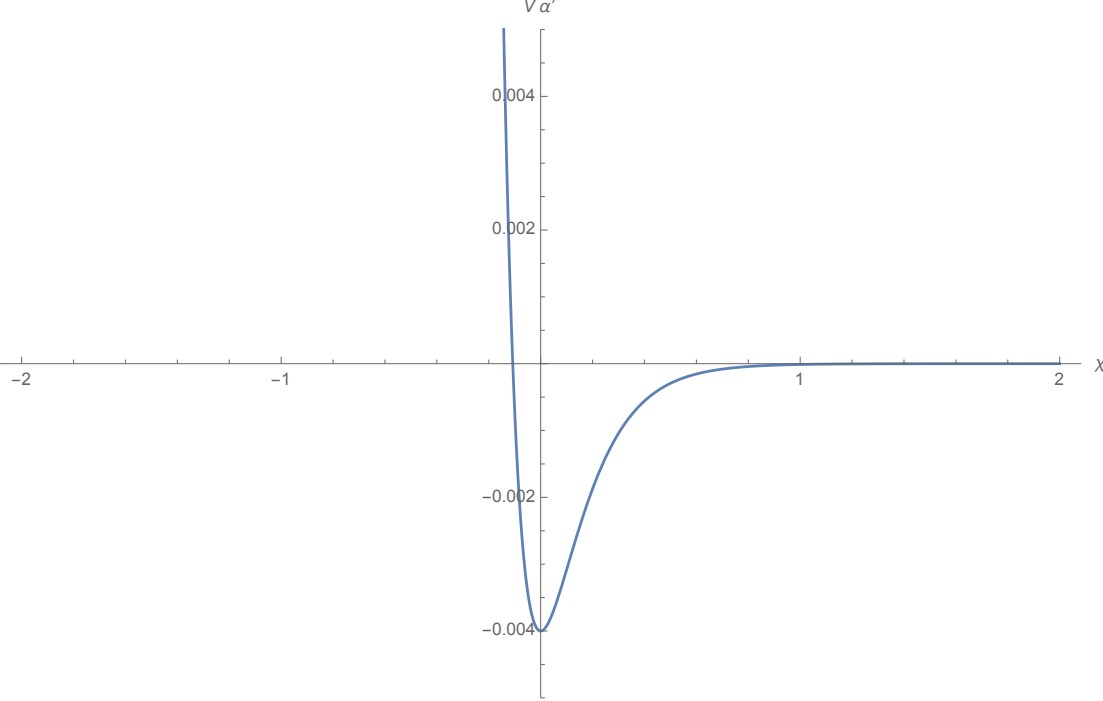

(b) The potential $V(\chi)$ with $\phi = \sigma = \hat{\chi} = 0$ and $n_5 = \hat{n}_5 = 10^3$.

Figure 1: The potential $V(\phi, \sigma, \chi, \hat{\chi})$ plotted with respect to its variables around its minimum.

$\hat{\chi} = 0$. This requirement fixes the values of $L_\circ, \hat{L}_\circ, g_{s,\circ}$. We use a linear combination of the derivatives to get cubic equations for $L_\circ^2$ and $\hat{L}_\circ^2$:

$$[3\partial_\phi + 2\partial_\chi]V_\circ\Big|_{\phi=\chi=\hat{\chi}=0} = 0\,, \tag{3.3}$$

$$\lambda\frac{g_{s,\circ}^2}{\alpha'}L_\circ^6 + 2L_\circ^4 - 2(\alpha'n_5)^2 = 0\,, \tag{3.4}$$

and

$$[3\partial_\phi + 2\partial_{\hat{\chi}}]V_\circ\Big|_{\phi=\chi=\hat{\chi}=0} = 0\,, \tag{3.5}$$

$$\lambda\frac{g_{s,\circ}^2}{\alpha'}\hat{L}_\circ^6 + 2\hat{L}_\circ^4 - 2(\alpha'\hat{n}_5)^2 = 0\,. \tag{3.6}$$

The solution of these cubic equations is

$$L_\circ^2 = \frac{2\alpha'}{3\lambda g_{s,\circ}^2}\left(\beta^{1/3} + \beta^{-1/3} - 1\right)\,, \tag{3.7}$$

$$\hat{L}_\circ^2 = \frac{2\alpha'}{3\lambda g_{s,\circ}^2}\left(\hat{\beta}^{1/3} + \hat{\beta}^{-1/3} - 1\right)\,, \tag{3.8}$$

where

$$\beta := -1 + \frac{27}{8}\lambda^2 n_5^2 g_{s,\circ}^4 + \frac{3}{8}\lambda|n_5|g_{s,\circ}^2\sqrt{D}, \qquad D := -48 + 81\lambda^2 n_5^2 g_{s,\circ}^4\,, \tag{3.9}$$

$$\hat{\beta} := -1 + \frac{27}{8}\lambda^2 \hat{n}_5^2 g_{s,\circ}^4 + \frac{3}{8}\lambda|\hat{n}_5|g_{s,\circ}^2\sqrt{\hat{D}}, \qquad \hat{D} := -48 + 81\lambda^2 \hat{n}_5^2 g_{s,\circ}^4\,. \tag{3.10}$$

When the discriminant is positive $(D > 0)$, the cube root $\beta^{1/3}$ is defined as the real root. The case when the discriminant is negative $(D < 0)$ is known as *casus irreducibilis*, and the solution for $L_\circ^2$ has to be irreducibly written in terms of complex numbers $\beta, \beta^{-1}$ even though the solution is a real number. To ensure the solution is positive in this case, we define the cube root $\beta^{1/3}$ as the root with the largest real part. The solution with respect to $n_1$ is plotted in figure 2a. Similar arguments apply for the hatted variables $\hat{L}_\circ^2, \hat{D}, \hat{\beta}$.

Let $N = \max(|n_5|, |\hat{n}_5|)$. We consider the small $g_{s,\circ}^2$ regime of $\lambda N g_{s,\circ}^2 \ll 1$ and get

$$L_\circ^2 = \alpha'|n_5|\left[1 - \frac{1}{4}\lambda|n_5|g_{s,\circ}^2 + \mathcal{O}(\lambda N g_{s,\circ}^2)^2\right]\,, \tag{3.11}$$

$$\hat{L}_\circ^2 = \alpha'|\hat{n}_5|\left[1 - \frac{1}{4}\lambda|\hat{n}_5|g_{s,\circ}^2 + \mathcal{O}(\lambda N g_{s,\circ}^2)^2\right]\,. \tag{3.12}$$

Next we compute the one-loop corrected string coupling. There is no closed-form expression for $g_{s,\circ}$. We show numerically the dependence of $g_{s,\circ}$ on $n_1$ for various $n_5, \hat{n}_5$ in

figure 2b. It is interesting that $g_{s,\circ}$ has an upper bound with respect to $n_1$, as opposed to tree-level $g_s$ which is unbounded analytically in the small $n_1$ limit; see figure 3. This shows that the non-supersymmetric theory avoids strong coupling, with the bound controlled by a function of $n_5, \hat{n}_5$. We derive this bound in the next section.

To obtain a closed-form expression, we use the small $g_s$ regime with $\lambda N g_s^2 \ll 1$. Using $\partial_\phi V_\circ \big|_{\phi=\chi=\hat{\chi}=0} = 0$, we get

$$g_{s,\circ}^2 = g_s^2 \left[ 1 - \frac{3}{8}\lambda \left( |n_5| + |\hat{n}_5| + \frac{L_{\text{AdS}}^2}{\alpha'} \right) g_s^2 + \mathcal{O}(\lambda N g_s^2)^2 \right] . \tag{3.13}$$

## 3.2 No-go for de Sitter

We compute the minimum of the total one-loop corrected potential giving,

$$V_\circ \big|_{\phi=\chi=\hat{\chi}=0} = \left( \frac{2(\alpha' n_5)^2}{L_\circ^6} - \frac{6}{L_\circ^2} + \frac{2(\alpha' \hat{n}_5)^2}{\hat{L}_\circ^6} - \frac{6}{\hat{L}_\circ^2} \right) + \frac{64\pi^2 \alpha'^6 g_{s,\circ}^4 n_1^2}{2r^2 L_\circ^6 \hat{L}_\circ^6} + 2\lambda \frac{g_{s,\circ}^2}{\alpha'} . \tag{3.14}$$

Using $n_1 = \frac{|n_5 \hat{n}_5|\sqrt{|n_5|+|\hat{n}_5|}}{4\pi g_s^2}$ and expanding $L_\circ, \hat{L}_\circ, g_{s,\circ}$ in terms of $g_s$ as in (3.11), (3.12), and (3.13), we get the small $g_s$ expansion

$$V_\circ \big|_{\phi=\chi=\hat{\chi}=0} = -\frac{2}{\alpha'} \left( \frac{1}{|n_5|} + \frac{1}{|\hat{n}_5|} \right) \left( 1 - \frac{1}{4}\lambda \frac{L_{\text{AdS}}^2}{\alpha'} g_s^2 + \mathcal{O}(\lambda N g_s^2)^2 \right) . \tag{3.15}$$

For de Sitter, we need a positive cosmological constant. Denote the one-loop corrected cosmological constant as

$$\Lambda_\circ := \frac{1}{2} V_\circ \big|_{\phi=\chi=\hat{\chi}=0} = \Lambda \left( 1 - \frac{1}{4}\lambda \frac{L_{\text{AdS}}^2}{\alpha'} g_s^2 + \mathcal{O}(\lambda N g_s^2)^2 \right) . \tag{3.16}$$

We see that to obtain de Sitter by flipping the sign of the cosmological constant, we would need to satisfy

$$\frac{1}{4} > \frac{1}{\lambda |n_5| g_s^2} + \frac{1}{\lambda |\hat{n}_5| g_s^2} . \tag{3.17}$$

However, this inequality is not satisfied in the $\lambda N g_s^2 \ll 1$ regime.

Therefore, we must turn to the numerical analysis to go beyond the small $g_s^2$ regime to check if de Sitter is possible. Figure 2c shows the cosmological constant for various $n_5, \hat{n}_5$ plotted against $n_1$. We see that the uplift to the cosmological constant is always small enough to ensure that it stays negative and we never get de Sitter. In particular, maximum uplift is obtained when $n_1 = 0$. We derive the uplifted cosmological constant for $n_1 = 0$ in the next section and show that it is never large enough to give a de Sitter solution.

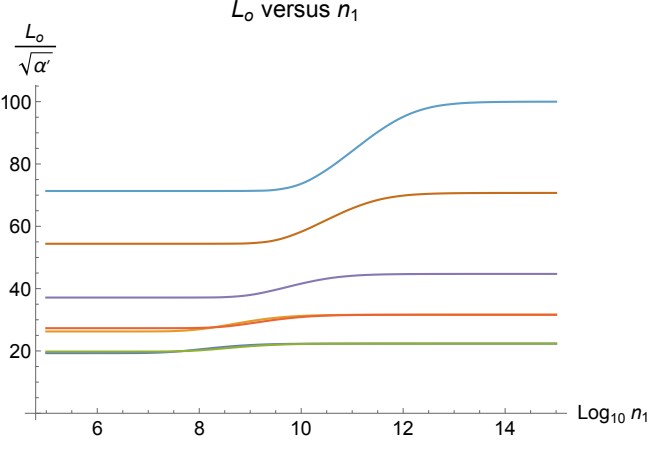

(a) One-loop corrected sphere size $L_\circ$ versus $n_1$.

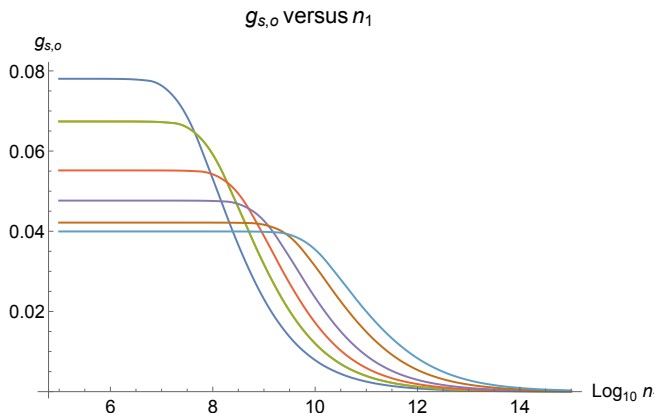

(b) One-loop corrected string coupling versus $n_1$.

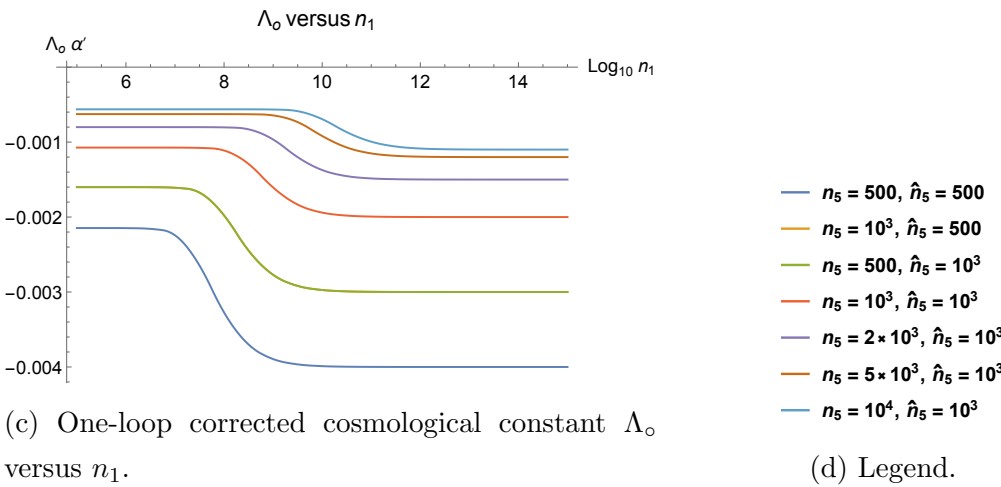

(c) One-loop corrected cosmological constant $\Lambda_\circ$ versus $n_1$.

(d) Legend.

Figure 2: Variables of interest plotted against $n_1$ for various $n_5, \hat{n}_5$. We see that all of them are sigmoid functions with respect to $n_1$.

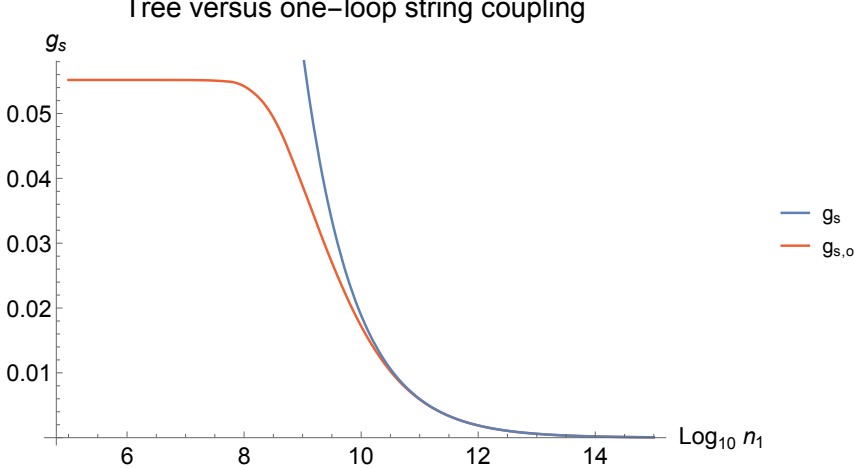

Figure 3: Comparison of tree level $g_s$ and one-loop corrected $g_{s,\circ}$ for $n_5 = \hat{n}_5 = 10^3$. Tree level $g_s$ diverges in the small $n_1$ limit, but $g_{s,\circ}$ is bounded.

## 4   Intrinsically quantum string vacua

When we set $n_1 = 0$, the AdS$_3$ background does not exist as a stable classical solution. However, the one-loop quantum correction (3.2) to the otherwise unstable classical potential stabilizes the background. The intuitive picture is that, classically, spacetime would collapse without an electric flux supporting it. In the case without supersymmetry, the one-loop quantum vacuum energy gives a potential energy contribution that prevents the AdS$_3$ spacetime from collapsing. As we discussed in the introduction, this is an *intrinsically quantum* string vacuum.

In section 3.1, we showed that there was no analytic solution for the one-loop corrected string coupling $g_{s,\circ}$ and sphere lengths $L_\circ, \hat{L}_\circ$ for generic $(n_1, n_5, \hat{n}_5)$. Fortunately, we have an analytical solution when $n_1 = 0$. The electric flux contribution to the potential vanishes in the potential

$$V_\circ\Big|_{\phi=\chi=\hat{\chi}=0} = \left( \frac{2(\alpha' n_5)^2}{L_\circ^6} - \frac{6}{L_\circ^2} + \frac{2(\alpha' \hat{n}_5)^2}{\hat{L}_\circ^6} - \frac{6}{\hat{L}_\circ^2} \right) + 2\lambda \frac{g_{s,\circ}^2}{\alpha'} . \tag{4.1}$$

We again require that the minimum is at $\phi = \chi = \hat{\chi} = 0$. The exact solution for $L_\circ, \hat{L}_\circ, g_{s,\circ}$ for any $n_5, \hat{n}_5$ is given in the supplementary Mathematica file and not here because of the size of the expression.

Let us consider the case $n_5 = \hat{n}_5 = n$ here, which significantly simplifies the expression.

We find

$$L_\circ^2 = \hat{L}_\circ^2 = \frac{\sqrt{5}}{3}\alpha'|n|\,, \tag{4.2}$$

$$g_{s,\circ}^2 = \frac{24}{5\sqrt{5}}\frac{1}{\lambda|n|}\,. \tag{4.3}$$

We see that the string coupling scales like $g_{s,\circ} \sim |n|^{-1/2}$. This is the upper bound of the string coupling $g_{s,\circ}$ with respect to $n_1$ for fixed $n_5 = \hat{n}_5 = n$,

$$\max_{n_1} g_{s,\circ}^2 = \frac{24}{5\sqrt{5}}\frac{1}{\lambda|n|} \approx \frac{3.04}{|n|}\,. \tag{4.4}$$

Remarkably, the non-supersymmetric theory avoids strong coupling with an upper bound controlled by $n_5, \hat{n}_5$.

Now we will show that de Sitter is not possible via a 1-loop uplift. For $n_5 = \hat{n}_5 = n$ the uplifted cosmological constant is

$$\Lambda_\circ = \frac{1}{2}V_\circ\Big|_{\phi=\chi=\hat{\chi}=0} = -\frac{12}{5\sqrt{5}}\frac{1}{\alpha'|n|}\,. \tag{4.5}$$

This is the maximum uplift one can get over all $n_1$. We conclude that it is impossible to uplift to de Sitter in this theory.

Finally, we can show that the theory is stable to all loop orders for large enough $n$. This is under the assumption that $g_s$ still controls a perturbative expansion for a background that is intrinsically quantum. In particular, solving for $g_s$ in the potential corrected to $m^{\text{th}}$ loop order will yield

$$\lambda g_{s,m}^2 + \lambda_2 g_{s,m}^3 + \cdots + \lambda_m g_{s,m}^{m+1} \propto \frac{1}{|n|}\,, \tag{4.6}$$

where $g_{s,m}$ is the $m$-loop corrected string coupling and $\lambda_i$ are dimensionless constants that can be computed by evaluating the $i^{th}$-loop vacuum amplitude. The left-hand side can be approximated by the leading term $\lambda g_{s,m}^2$ to arbitrary precision for small enough $g_{s,m}$. At the same time, $n$ can be chosen large enough so that the right-hand side falls within the range in which $\lambda g_{s,m}^2$ is a good approximation. Therefore, the 1-loop solution agrees with the $m$-loop solution to arbitrary precision for large enough $n$.

## 5 Stability and spectrum analysis

To determine the perturbative stability of the theory for all $(n_1, n_5, \hat{n}_5)$ values, we will derive the masses of the scalars in the theory. We have already argued that all the massless

modes which correspond to single trace operators are free of tadpoles. There are, however, tachyonic modes in this compactification. We will need to show that the tachyons in our model are above the Breitenlohner-Freedman (BF) bound [35], which is a requirement of perturbative stability.

Unlike flat space, tachyons in $\text{AdS}_{d+1}$ do not necessarily signal instability. As long as a tachyon has mass squared above the BF bound,

$$m^2 \geq m^2_{\text{BF}} := -\frac{d^2}{4L^2_{\text{AdS}}} \,, \tag{5.1}$$

it does not generate an instability. This is because the gravitational potential gives tachyons above the BF bound an overall positive energy. We show in this section that all the scalars of the $O(16) \times O(16)$ heterotic string on $\text{AdS}_3 \times S^3 \times S^3 \times S^1$ have mass squared above the BF bound, which in this dimension is

$$m^2_{BF} = -L^{-2}_{\text{AdS}} \,. \tag{5.2}$$

For our model, we need to take quantum corrections to the vacuum energy into account and use the one-loop corrected AdS length scale $L_{\text{AdS},\circ}$ to determine the one-loop corrected BF bound:

$$m^2_{BF,\circ} := -L^{-2}_{\text{AdS},\circ} \,. \tag{5.3}$$

Another possible threat to perturbative stability is the existence of multi-trace marginal operators. Such operators might develop tadpoles which would destabilize the theory. From a spacetime perspective, they correspond to multi-particle states with precisely the right masses to correspond to a marginal operator in a putative holographic description. In section 5.6, we show that multi-particle states corresponding to such marginal operators do not exist in this theory for most choices of quantization.

This section is organized as follows: in section 5.1 we first consider the effective theory of the $O(16) \times O(16)$ heterotic string in string frame reduced on $S^1$ together with the one-loop potential, and we solve for the loop-corrected AdS length scale $L_{\text{AdS},\circ}$. In section 5.2, we decompose the fluctuations of the spacetime fields. In section 5.3, we consider the action expanded to quadratic order in these fluctuations. In section 5.4, we present the derivation of the free scalar spectrum of the theory, which is identical to the supersymmetric case in [38, section 3.4]. In section 5.5, we show that there are no scalars with mass below the BF bound. Multi-trace operators are examined in section 5.6 with dangerous operators seemingly absent for most choices of quantization. Consequently, we conclude that this model is perturbatively stable.

## 5.1 Vacuum solution in string frame

The effective action in the massless bosonic sector of the $O(16) \times O(16)$ heterotic string is identical to that of the NS-NS sector of the type IIB string, except for the appearance of a non-abelian gauge field in the heterotic case. The spectrum of type IIB string theory on $\mathrm{AdS}_3 \times S_3 \times \hat{S}_3 \times S_1$ has been derived in detail in [38]. Up until section 5.5.2, what follows should be treated as a review of [38] with modifications due to the terms associated to the non-abelian gauge field $\mathcal{A}$ and to the one-loop potential generated by the broken supersymmetry.

The $D = 10$ bosonic action is

$$S_{D=10} = \frac{1}{2\kappa_{10}^2} \int d^{10}x \sqrt{-g} e^{-2\phi} \left( R + 4(\partial\phi)^2 - \frac{1}{2 \times 3!}|H_3|^2 - \frac{1}{4}|\mathcal{F}|^2 \right), \tag{5.4}$$

where $\mathcal{F}$ is the field strength for $\mathcal{A}$. Next, we compactify on the circle $S^1$ and dimensionally reduce the fields by defining $A_\mu := g_{\mu,10}$, $\hat{A}_\mu := B_{\mu,10}$, with field strengths $F, \hat{F}$ respectively. We also define scalars $\varphi^\alpha = \mathcal{A}_{10}{}^\alpha$ transforming as $(\mathbf{120}, \mathbf{1}) \oplus (\mathbf{1}, \mathbf{120})$ under $\mathrm{SO}(16) \times \mathrm{SO}(16)$, where $\alpha$ is raised and lowered by the Killing form $\kappa_{\alpha\beta}$. Finally, we add the one-loop potential (B.27) to quadratic order in the fluctuations. See also [41, section 3.2] for details about the dimensional reduction of heterotic theories. The action then becomes:

$$\begin{aligned} S_{D=9} = \frac{1}{2\kappa_9^2} \int d^9x \sqrt{-g} \Big( e^{-2\phi} \Big\{ R + 4(\partial\phi)^2 - \frac{1}{2 \times 3!}|H_3|^2 - (\partial\sigma)^2 - \frac{1}{2}(D_\mu\varphi^\alpha)(D^\mu\varphi_\alpha) \\ - \frac{1}{4}|F|^2 - \frac{1}{4}|\hat{F}|^2 - \frac{1}{4}|\mathcal{F}|^2 \Big\} - 2\lambda\frac{g_{s,\circ}^2}{\alpha'} - m_\sigma^2\sigma^2 - \frac{1}{2}m_\alpha^2(\varphi^\alpha)^2 \Big). \end{aligned} \tag{5.5}$$

The mass terms for $\sigma$ and the Wilson line moduli $\varphi^\alpha$ come from the fact that the potential is at a minimum for these moduli when the radius of the $S^1$ is chosen to be the self-dual radius. We will look for vacuum solutions with $\phi = \sigma = \varphi^a = 0$ and $F = \hat{F} = \mathcal{F} = 0$.[3]

The equation of motion for $H_3$ is

$$d * H_3 = 0. \tag{5.6}$$

The dilaton equation of motion gives

$$R - \frac{1}{2 \cdot 3!}|H_3|^2 = 4(\partial\phi)^2 + 4\Box\phi. \tag{5.7}$$

Setting $\phi = 0$ gives the following relationship between the Ricci curvature and flux,

$$R = \frac{1}{2 \cdot 3!}|H_3|^2. \tag{5.8}$$

---

[3]Note that since we are in the supergravity regime with large $n_5$ and $\hat{n}_5$, stringy $\alpha'$ corrections in (2.3) to $H_3$ from the choice of the background gauge field are negligible.

Next we consider the Einstein equation

$$R_{\mu\nu} - \frac{1}{4}H_{\mu\rho\sigma}H_\nu{}^{\rho\sigma} + \left(\lambda\frac{g_{s,\circ}^2}{\alpha'}\right)g_{\mu\nu} = 0 \,. \tag{5.9}$$

Taking the trace gives

$$R - \frac{1}{4}|H_3|^2 + 9\lambda\frac{g_{s,\circ}^2}{\alpha'} = 0 \,. \tag{5.10}$$

Using (5.8) and (5.10), we get

$$R = \frac{1}{2\cdot 3!}|H_3|^2 = \frac{9}{2}\frac{\lambda g_{s,\circ}^2}{\alpha'} \,. \tag{5.11}$$

We also see that

$$R_{\mu\nu} = -\frac{2}{L_{\text{AdS},\circ}^2}g_{\mu\nu}^{\text{AdS}} + \frac{2}{L_\circ^2}g_{\mu\nu}^{S^3} + \frac{2}{\hat{L}_\circ^2}g_{\mu\nu}^{\hat{S}^3} \,, \tag{5.12}$$

$$R = -\frac{6}{L_{\text{AdS},\circ}^2} + \frac{6}{L_\circ^2} + \frac{6}{\hat{L}_\circ^2} \,, \tag{5.13}$$

where $g_{\mu\nu}^{\mathcal{M}}$ is simply $g_{\mu\nu}$ when $\mu, \nu$ are both indices in $\mathcal{M}$ and zero otherwise. From (5.11) and (5.13), we get the expression for the AdS length scale:

$$\begin{aligned}
L_{\text{AdS},\circ}^{-2} &= \frac{1}{L_\circ^2} + \frac{1}{\hat{L}_\circ^2} - \frac{3}{4}\lambda\frac{g_{s,\circ}^2}{\alpha'} \\
&= L_{\text{AdS}}^{-2}\left(1 - \frac{1}{4}\lambda\frac{L_{\text{AdS}}^2}{\alpha'}g_s^2 + \mathcal{O}(\lambda N g_s^2)\right) \,.
\end{aligned} \tag{5.14}$$

We write the components of $H_3$ as

$$H_3 = \frac{2}{\mathfrak{L}_{\text{AdS},\circ}}\epsilon_3 + \frac{2}{\mathfrak{L}_\circ}\epsilon_{S^3} + \frac{2}{\hat{\mathfrak{L}}_\circ}\epsilon_{S^3} \,. \tag{5.15}$$

Here, we introduced the length scales $\mathfrak{L}_{\text{AdS},\circ}, \mathfrak{L}_\circ, \hat{\mathfrak{L}}_\circ$ associated to the fluxes through the spaces $\text{AdS}_3, S^3, \hat{S}^3$, respectively. From flux quantization we get

$$H_3 = \frac{8\pi\alpha'^3 g_{s,\circ}^2 n_1}{rL_\circ^3\hat{L}_\circ^3}\epsilon_3 + \frac{2\alpha' n_5}{L_\circ^3}\epsilon_{S^3} + \frac{2\alpha'\hat{n}_5}{L_\circ^3}\epsilon_{\hat{S}^3} \,, \tag{5.16}$$

so that the flux length scales can be written in terms of the length scales of the spaces:

$$\mathfrak{L}_{\text{AdS},\circ} = \frac{rL_\circ^3\hat{L}_\circ^3}{4\pi\alpha'^3 g_{s,\circ}^2 n_1} \,, \tag{5.17}$$

$$\mathfrak{L}_\circ = L_\circ\frac{L_\circ^2}{\alpha' n_5} \,, \tag{5.18}$$

$$\hat{\mathfrak{L}}_\circ = \hat{L}_\circ\frac{\hat{L}_\circ^2}{\alpha' n_5} \,. \tag{5.19}$$

Note that at tree level, the flux and space length scales coincide

$$\mathfrak{L}_{\text{AdS}} = L_{\text{AdS}} \,, \qquad \mathfrak{L} = L \,, \qquad \hat{\mathfrak{L}} = \hat{L} \,. \tag{5.20}$$

It follows from the components of the Einstein equation (5.9) that

$$-\frac{2}{L_{\text{AdS,o}}^2} + \frac{2}{\mathfrak{L}_{\text{AdS,o}}^2} + \lambda \frac{g_{s,\text{o}}^2}{\alpha'} = 0 \,, \tag{5.21}$$

$$\frac{2}{L_{\text{o}}^2} - \frac{2}{\mathfrak{L}_{\text{o}}^2} + \lambda \frac{g_{s,\text{o}}^2}{\alpha'} = 0 \,, \tag{5.22}$$

$$\frac{2}{\hat{L}_{\text{o}}^2} - \frac{2}{\hat{\mathfrak{L}}_{\text{o}}^2} + \lambda \frac{g_{s,\text{o}}^2}{\alpha'} = 0 \,. \tag{5.23}$$

These equations are exactly the ones that we solved in the $d = 3$ Einstein frame, as expected. We conclude that the $d = 9$ string frame solution is equivalent to the $d = 3$ Einstein frame solution of section 3.1.

## 5.2 Decomposition of fields

Our conventions for the decomposition of the field fluctuations follows [38]. We decompose the components of the fluctuations in the metric as

$$\delta g_{\mu\nu} = H_{\mu\nu} + g_{\mu\nu}M \,, \quad g^{\mu\nu}H_{\mu\nu} = 0 \,, \tag{5.24}$$

$$\delta g_{\mu a} = R_{\mu a} \,, \tag{5.25}$$

$$\delta g_{\mu i} = S_{\mu i} \,, \tag{5.26}$$

$$\delta g_{ai} = T_{ai} \,, \tag{5.27}$$

$$\delta g_{ab} = K_{ab} + g_{ab}N \,, \quad g^{ab}K_{ab} = 0 \,, \tag{5.28}$$

$$\delta g_{ij} = L_{ij} + g_{ij}P \,, \quad g^{ij}L_{ij} = 0 \,. \tag{5.29}$$

Here and subsequently, greek indices refer to AdS$_3$, latin indices from the beginning of the alphabet $a, b, \dots$ refer to $S^3$, and the latin indices from the middle of the alphabet $i, j, \dots$ refer to $\hat{S}^3$. Capital latin letters $M, N, \dots$ refer to the 9 dimensional indices.

We decompose the components of the fluctuations in the antisymmetric tensor field

$\delta B = X$ as

$$X_{\mu\nu} = \epsilon_{\mu\nu\rho}U^\rho \ , \tag{5.30}$$

$$X_{ab} = \epsilon_{abc}V^c \ , \tag{5.31}$$

$$X_{ij} = \epsilon_{ijk}W^k \ , \tag{5.32}$$

$$X_{\mu a} = C_{\mu a} \ , \tag{5.33}$$

$$X_{\mu i} = D_{\mu i} \ , \tag{5.34}$$

$$X_{ai} = E_{ai} \ . \tag{5.35}$$

The basis for 0-forms on $S^3$ is given by

$$Y^{(\ell,0)} \ . \tag{5.36}$$

The basis for 1-forms on $S^3$ is given by

$$Y_a^{(\ell,\pm 1)} \ , \quad \partial_a Y^{(\ell,0)} \ . \tag{5.37}$$

The space of traceless symmetric 2-tensors is spanned by

$$Y_{ab}^{(\ell,\pm 2)} \ , \quad \nabla_{\{a} Y_{b\}}^{(\ell,\pm 1)} \ , \quad \nabla_{\{a}\nabla_{b\}} Y^{(\ell,0)} \ . \tag{5.38}$$

Here, $\{ab\}$ denotes the traceless symmetric part, and $Y_{(s)}^{(\ell_1,\ell_2)}$ and $\hat{Y}_{(s)}^{(\hat{\ell}_1,\hat{\ell}_2)}$ are the eigenfunctions of the Laplacian on $S^3$ and $\hat{S}^3$ transforming under $SO(4) \cong SU(2) \times SU(2)$ in the representation

$$\frac{1}{2}(\ell_1 + \ell_2) \ , \quad \frac{1}{2}(\ell_1 - \ell_2) \ . \tag{5.39}$$

In terms of these bases, arbitrary fields of spin $0, 1, 2$ on $S^3$ and spin $0$ on $\hat{S}^3$ are decomposed as follows:

$$\phi = \sum_{\ell,\hat{\ell}} \phi^{(\ell,0)(\hat{\ell},0)} \, Y^{(\ell,0)} \hat{Y}^{(\hat{\ell},0)} \ , \tag{5.40}$$

$$V_a = \sum_{\ell,\hat{\ell}} V^{(\ell,\pm 1)(\hat{\ell},0)} \, Y_a^{(\ell,\pm 1)} \hat{Y}^{(\hat{\ell},0)} + V^{(\ell,0)(\hat{\ell},0)} \, \partial_a Y^{(\ell,0)} \hat{Y}^{(\hat{\ell},0)} \ , \tag{5.41}$$

$$K_{ab} = \sum_{\ell,\hat{\ell}} K^{(\ell,\pm 2)(\hat{\ell},0)} \, Y_{ab}^{(\ell,\pm 2)} \hat{Y}^{(\hat{\ell},0)} + K^{(\ell,\pm 1)(\hat{\ell},0)} \, \nabla_{\{a} Y_{b\}}^{(\ell,\pm 1)} \hat{Y}^{(\hat{\ell},0)}$$
$$+ K^{(\ell,0)(\hat{\ell},0)} \, \nabla_{\{a}\nabla_{b\}} Y^{(\ell,0)} \hat{Y}^{(\hat{\ell},0)} \ . \tag{5.42}$$

The notation $\{ab\}$ denotes the symmetric traceless combination of indices.

We choose the Lorentz gauge as in [38, 42] ,

$$\nabla^a h_{a\mu} = 0 \ , \quad \nabla^a h_{\{ab\}} = 0 \ , \quad \nabla^a h_{ai} = 0 \ , \quad \nabla^a X_{aM} = 0 \ . \tag{5.43}$$

This gauge choice breaks the manifest symmetry between the two spheres, but it turns out the spectrum is still symmetric with respect to the exchange of the two spheres as it should be.

## 5.3 Fluctuating the equations of motion

The action to quadratic order in fluctuations is

$$\delta^2 S_{D=9} = \frac{1}{4\kappa_9^2} \int d^9 x \sqrt{-g} \mathcal{L} \,, \tag{5.44}$$

where

$$
\begin{aligned}
\mathcal{L} = & \frac{1}{2} \nabla_P h^R{}_R \nabla^P h^M{}_M - \frac{1}{2} \nabla_R h_{MP} \nabla^R h^{MP} + \nabla_P h_{MR} \nabla^R h^{MP} - \nabla_P h^M{}_M \nabla^R h^P{}_R \\
& - \frac{1}{2} H_{MR}{}^N H_{PQN} h^{MP} h^{RQ} - 8 \nabla_P \phi \nabla^{[P} h^{M]}{}_M + 8 \nabla_M \phi \nabla^M \phi - 2 \nabla_M \sigma \nabla^M \sigma \\
& + \frac{1}{6} H_{MPR} \Xi^{MPR} (4\phi - h^Q{}_Q) + H_M{}^{RQ} h^{MP} \Xi_{PRQ} - \frac{1}{6} \Xi_{MPR} \Xi^{MPR} \\
& - \frac{1}{2} Z_{MP} Z^{MP} - \frac{1}{2} \hat{Z}_{MP} \hat{Z}^{MP} - \frac{1}{2} \mathcal{Z}_{MP} \mathcal{Z}^{MP} - \nabla_M \varphi \nabla^M \varphi \\
& + \lambda \frac{g_{s,\circ}^2}{\alpha'} \left( \frac{1}{2} (h^M{}_M)^2 - h_{MP} h^{MP} - 4 h^M{}_M \phi \right) . \tag{5.45}
\end{aligned}
$$

Here, $\phi$ is the fluctuation in the dilaton field, $\varphi$ is the fluctuation in the scalars from the gauge fields, $\sigma$ is the fluctuation in the size of the circle,[4] $\delta H_{MNP} = \Xi_{MNP}$, $\delta g_{MN} = h_{MN}$, $\delta F_{MP} = Z_{MP}$, $\delta \hat{F}_{MP} = \hat{Z}_{MP}$, and $\delta \mathcal{F}_{MP} = \mathcal{Z}_{MP}$. We have omitted the mass terms for the $\sigma$ and $\varphi$ fields.

Note that (5.45) is obtained by expanding the action (5.5) around the loop-corrected background solution. So, although (5.5) does not have any dilaton dependence in the one-loop terms, we will generate terms that involve $\phi$ and are proportional to $\lambda g_{s,\circ}^2$ (i.e. the last term in (5.45)) when we expand in fluctuations.

From the action we can see that $\sigma, \varphi, Z_{MP}, \hat{Z}_{MP}, \mathcal{Z}_{MP}$ are free fields. Their mass spectrum will be computed in the next section straightforwardly. The computation of the spectrum of $h_{MP}, \phi, \Xi_{MPR}$ is more involved and will be explained in section 5.5, with the details provided in Appendix C.

## 5.4 Free scalar spectrum

We first consider the free scalars $\sigma, \varphi^\alpha$. Without loss of generality, we will do the computations for $\sigma$, but they apply exactly to $\varphi$ as well. The equation of motion is

$$(\Box + m_\sigma^2)\sigma = 0 \,. \tag{5.46}$$

---

[4]We abuse notation somewhat by denoting the fluctuation of these fields by the same symbols we used for the fields themselves. However, there is no real ambiguity since these fields themselves do not show up in the equations anymore—only the fluctuations around a zero expectation value will appear.

Expanding the modes and considering the components, we get a Klein-Gordon equation in AdS$_3$

$$(\Box_0 + \Box_x + \Box_{\hat{x}} + m_\sigma^2)\sigma^{(\ell,0)(\hat{\ell},0)} = (\Box_0 + m_{\ell,\hat{\ell}}^2 + m_\sigma^2)\sigma^{(\ell,0)(\hat{\ell},0)} = 0 \,, \tag{5.47}$$

where

$$m_{\ell,\hat{\ell}}^2 := \frac{\ell(\ell+2)}{L^2} + \frac{\hat{\ell}(\hat{\ell}+2)}{\hat{L}^2} \,, \qquad \ell, \hat{\ell} \in \mathbb{Z}_{\geq 0} \,. \tag{5.48}$$

Next, we have the gauge field perturbations $Z_{MP}, \hat{Z}_{MP}, \mathcal{Z}_{MP}$, collectively with similar equations of motion. Without loss of generality, we only consider $Z_{MP}$, but the analysis applies exactly to $\hat{Z}_{MP}$ and $\mathcal{Z}_{MP}$ as well.

We denote the perturbation of the gauge field by $\delta A_M = \Upsilon_M$ and decompose it as follows,

$$\Upsilon_\mu = \Pi_\mu \,, \quad \Upsilon_a = \Sigma_a \,, \quad \Upsilon_i = \Omega_i \,. \tag{5.49}$$

We again use the Lorentz gauge

$$\nabla^a \Sigma_a = 0 \,. \tag{5.50}$$

The equations of motion are

$$0 = (\Box_0 + \Box_x + \Box_{\hat{x}})\Pi_\mu - \nabla_\mu \nabla^\nu \Pi_\nu - \nabla_\mu \nabla^i \Omega_i \,, \tag{5.51}$$

$$0 = (\Box_0 + \Box_x + \Box_{\hat{x}})\Sigma_a - \nabla_a \nabla^\nu \Pi_\nu - \nabla_a \nabla^i \Omega_i \,, \tag{5.52}$$

$$0 = (\Box_0 + \Box_x + \Box_{\hat{x}})\Omega_i - \nabla_i \nabla^\nu \Pi_\nu - \nabla_i \nabla^j \Omega_j \,, \tag{5.53}$$

modulo positive mass squared terms due to one-loop. We now consider scalar modes with respect to AdS$_3$ as well as $S^3$ and $\hat{S}^3$. There are two such scalars: $\nabla^\mu \Pi_\mu$ and $\nabla^i \Omega_i$. Thus, we have an overconstrained system, since the scalar parts of (5.51) – (5.53) constitute three equations for these two fields. We extract the scalar parts of the three equations by applying $\nabla^\mu$ to the first equation, $\nabla^a$ to the second equation, and $\nabla^i$ to the third equation. We get

$$0 = (\Box_x + \Box_{\hat{x}})\nabla^\mu \Pi_\mu - \Box_0 \nabla^i \Omega_i \,, \tag{5.54}$$

$$0 = \Box_x (\nabla^\nu \Pi_\nu + \nabla^i \Omega_i) \,, \tag{5.55}$$

$$0 = (\Box_0 + \Box_x)\nabla^\mu \Omega_i - \Box_{\hat{x}}\nabla^\mu \Pi_\mu \,. \tag{5.56}$$

For $\ell > 0$, the second equation implies that there is actually only one scalar field, which we take to be $\nabla^\mu \Pi_\mu$. The first and third equation are equivalent to

$$0 = (\Box_0 + \Box_x + \Box_{\hat{x}})\nabla^\mu \Pi_\mu = (\Box_0 + m_{\ell,\hat{\ell}}^2)\nabla^\mu \Pi_\mu \,. \tag{5.57}$$

We again get scalars with their masses corrected by $+m_{\ell,\hat{\ell}}^2$. When $\ell = 0$, there are no scalars as shown in [38, Appendix C.3].

## 5.5 Coupled scalar spectrum

The analysis of the remaining fluctuations is more complicated. Each equation of motion involves both the spacetime and the internal Laplacian acting on a number of fields, as well as linear combinations of the fields themselves. So all the fields mix, but one can diagonalize the corresponding mass matrix. The eigenvalues correspond to the masses of the particles, since the system of equations is simply the Klein-Gordon equation of seven coupled scalar particles on $\text{AdS}_3$.

We have relegated most of the details to Appendix C.

### 5.5.1 Tree level

We first consider the tree-level spectrum. This means that we take $\lambda = 0$. In this case, the eigenvalues of the mass matrix, and thus the mass squared, of the seven coupled scalars is [38, section 3.5]

$$\lambda_1 = m^2_{\ell,\hat{\ell}-2}, \tag{5.58}$$

$$\lambda_2 = m^2_{\ell-2,\hat{\ell}}, \tag{5.59}$$

$$\lambda_3 = m^2_{\ell,\hat{\ell}}, \tag{5.60}$$

$$\lambda_4 = m^2_{\ell,\hat{\ell}+2}, \tag{5.61}$$

$$\lambda_5 = m^2_{\ell+2,\hat{\ell}}, \tag{5.62}$$

$$\lambda_6 = \left(2L_{\text{AdS}}^{-1} + \sqrt{L_{\text{AdS}}^{-2} + m^2_{\ell,\hat{\ell}}}\right)^2 - L_{\text{AdS}}^{-2}, \tag{5.63}$$

$$\lambda_7 = \left(2L_{\text{AdS}}^{-1} - \sqrt{L_{\text{AdS}}^{-2} + m^2_{\ell,\hat{\ell}}}\right)^2 - L_{\text{AdS}}^{-2}. \tag{5.64}$$

When $\ell \leq 1$ or $\hat{\ell} \leq 1$, some of the scalars are gauged away. The remaining scalars are listed in Appendix D. It follows that only $\lambda_7$ can correspond to a tachyon. In fact, $\lambda_7$ saturates the BF bound for $\ell = \hat{\ell} = 1$,

$$\lambda_{7,\ell=\hat{\ell}=1} = -L_{\text{AdS}}^{-2} = m^2_{BF}. \tag{5.65}$$

### 5.5.2 One-loop level for small $g_s$

For large $n_1$ (small $g_s$), $\lambda_7$ is the only mode that can threaten perturbative stability because it sits right at the BF bound. In the small $g_s$ regime of $\lambda N g_s^2 \ll 1$, this scalar has mass

Figure 4: We fix $n_5 = \hat{n}_5 = 10^3$. Mass squared of the BF bound saturating tachyon $\lambda_{7,\ell=\hat{\ell}=1,\circ}$ plotted against $n_1$. The tachyon starts out at the BF bound in the large $n_1$ limit and masses up with decreasing $n_1$.

given by

$$\lambda_{7,\ell=\hat{\ell}=1,\circ}L^2_{\text{AdS},\circ} = -1 + \lambda\frac{L^2_{\text{AdS}}}{\alpha'}g_s^2 + \mathcal{O}(\lambda N g_s^2)^2\,. \tag{5.66}$$

Here, we express the mass squared of the scalar with respect to the loop-corrected AdS length scale $L^2_{\text{AdS},\circ}$ on the left-hand side. Using the one-loop corrected BF bound, we see that

$$\lambda_{7,\ell=\hat{\ell}=1,\circ} \geq m^2_{BF,\circ} = -L^{-2}_{\text{AdS},\circ}\,. \tag{5.67}$$

So we conclude that the model is perturbatively stable for small $g_s^2$ pending an investigation of multi-trace operators to which we will turn shortly. For completeness, we provide the small $g_s$ correction terms for other scalars with low $\ell, \hat{\ell}$ in Appendix D.1.

### 5.5.3 One-loop level for all $g_s$

For non-negligible $g_s^2$, solving for the eigenvalues of the mass matrix does not yield closed form expressions. Therefore we rely on numerical results. The tachyon that saturates the BF bound masses up in the large $g_s^2$ limit as shown in figure 4.

In fact, all scalar masses are sigmoid functions with respect to $g_s^2$, i.e. they monotonically transition between their asymptotic values at small and large $g_s^2$. Therefore, if the first order

correction to the mass squared is positive, then the scalar will mass up and vice versa. With this in mind, we can see that since $\lambda_{7,\ell=\hat{\ell}=1,\circ}$ receives a positive correction, it will always gain mass.

There are other scalars that do receive negative first order corrections. We show that they do not cross the BF bound in the large $g_s^2$ limit in Appendix D.2. Using the fact that all of the scalar masses have a transition function similar to figure 4 and that there are no BF bound violations for either asymptote, we conclude that there are no scalars below the BF bound for any value of $(n_1, n_5, \hat{n}_5)$, and so the model is always perturbatively stable with respect to single particle excitations.

## 5.6 Marginal multi-trace operators

Marginal multi-trace operators correspond to multi-particle excitations whose dual conformal dimensions sum up to 2. For a field of mass squared $m^2 L_{\text{AdS},\circ}^2 \geq -\frac{d^2}{4} + 1$ in $\text{AdS}_{d+1}$, the total conformal dimension of the dual operator is given by

$$\Delta = \frac{d}{2} + \sqrt{\frac{d^2}{4} + m^2 L_{\text{AdS},\circ}^2} \,. \tag{5.68}$$

Tachyons above the BF bound with mass squared in the range

$$-\frac{d^2}{4} < m^2 L_{\text{AdS},\circ}^2 < -\frac{d^2}{4} + 1 \tag{5.69}$$

fall within the window discussed by Klebanov and Witten [43]. These tachyons admit two distinct quantizations in AdS and two corresponding dual interpretations. The two choices of conformal dimension that can be assigned to the dual operator are

$$\Delta_\pm = \frac{d}{2} \pm \sqrt{\frac{d^2}{4} + m^2 L_{\text{AdS},\circ}^2} \,. \tag{5.70}$$

For our $d = 2$ case massive fields have conformal dimension greater than 2, massless fields have $\Delta = 2$, and tachyons above the BF bound have $0 < \Delta < 2$. Therefore, we only need to check the possible conformal dimensions of multi-particle states constructed from the tachyons.

*The spectrum of tachyons*

We consider the case of small $g_s^2$. As shown in Appendix D, there are three possible tachyons. The first order corrections to their mass squared are given by

$$(\ell = 1, \hat{\ell} = 1): \qquad \lambda_{7,\circ} L^2_{\text{AdS},\circ} = -1 + 2\lambda \frac{L^2_{\text{AdS}}}{\alpha'} g_s^2 + \mathcal{O}(\lambda N g_s^2)^2 \,, \tag{5.71}$$

$$(\ell = 2, \hat{\ell} = 0): \qquad \lambda_{2,\circ} L^2_{\text{AdS},\circ} = 0 - \frac{1}{3}\lambda \frac{L^2_{\text{AdS}}}{\alpha'} g_s^2 + \mathcal{O}(\lambda N g_s^2)^2 \,, \tag{5.72}$$

$$(\ell = 0, \hat{\ell} = 2): \qquad \lambda_{2,\circ} L^2_{\text{AdS},\circ} = 0 - \frac{1}{3}\lambda \frac{L^2_{\text{AdS}}}{\alpha'} g_s^2 + \mathcal{O}(\lambda N g_s^2)^2 \,. \tag{5.73}$$

Their corresponding conformal dimensions are

$$\Delta^{(1,1),7}_{\pm} = 1 \pm \left( \sqrt{\frac{2\lambda}{\alpha'}} L_{\text{AdS}} g_s + \mathcal{O}(\lambda N g_s^2) \right) \,, \tag{5.74}$$

$$\Delta^{(2,0),2}_{\pm} = 1 \pm \left( 1 - \frac{1}{6}\lambda \frac{L^2_{\text{AdS}}}{\alpha'} g_s^2 + \mathcal{O}(\lambda N g_s^2)^2 \right) \,, \tag{5.75}$$

$$\Delta^{(0,2),2}_{\pm} = 1 \pm \left( 1 - \frac{1}{6}\lambda \frac{L^2_{\text{AdS}}}{\alpha'} g_s^2 + \mathcal{O}(\lambda N g_s^2)^2 \right) \,, \tag{5.76}$$

where $\Delta^{(\ell,\hat{\ell}),k}$ denotes the conformal dimension of the $\lambda_k$ scalar at the given values of $\ell, \hat{\ell}$. We see that if the $(\ell = 2, \hat{\ell} = 0)$ and $(\ell = 0, \hat{\ell} = 2)$ modes are quantized with opposite signs, we do get a marginal double trace operator. With this assignment of conformal dimension, a deeper investigation is required to determine whether the marginal operator develops a beta function. If any other assignment is chosen, there are no marginal multi-trace operators for generic $(n_1, n_5, \hat{n}_5)$, since there is no exact cancellation of the correction to the conformal dimensions of the scalars.

This argument extends to the large tree-level $g_s$ case. There are only two tachyons, one with $(\ell = 2, \hat{\ell} = 0)$ and another with $(\ell = 0, \hat{\ell} = 2)$. Their masses are identical because of the symmetry of the spectrum with respect to the exchange of the two spheres $S^3 \leftrightarrow \hat{S}^3$. At $n_1 = 0$, their masses are given in figure 8. Again, choosing opposite signs in the assignment of conformal dimension would result in a marginal double trace operator. However, because there no other tachyons, this is the only such operator and it is not in the theory for any other choice of quantization. In particular, for generic $(n_1, n_5, \hat{n}_5)$ with quantizations of the $(\ell = 2, \hat{\ell} = 0)$ and $(\ell = 0, \hat{\ell} = 2)$ tachyons chosen with the same sign, there are no marginal multi-trace operators.

*An argument from gauge invariance*

There is a second way to argue that marginal multi-trace operators are not problematic in these backgrounds, even for the choice of quantization that might allow such an operator. Notice that all the tachyons described in the preceeding discussion are charged under the gauge symmetry generated from the isometries of $S^3$ and $\hat{S}^3$. The product of tachyons needed to generate a marginal multi-trace operator involves one excitation of (5.72) and one excitation of (5.73) quantized with opposite signs. However, this is a charged bilinear in spacetime which is protected from tadpoles by gauge symmetry [26]. This argument really requires very little detailed information about the mass spectrum, but implies that multi-trace operators are not problematic with any choice of quantization in AdS$_3$.

## Acknowledgements

We would like to thank Ivano Basile, Lorenz Eberhardt, Simone Giombi, Jeff Harvey, Igor Klebanov, David Kutasov, Finn Larsen, Emanuel Malek, Eric Perlmutter and Augusto Sagnotti for helpful discussions. Z. B. is supported in part by the Purcell fellowship and the James Mills Peirce fellowship at Harvard University and in part by NSF Grant No. PHY2014195. S. S. is supported in part by NSF Grant No. PHY2014195. D. R. is supported in part by NSF Grant No. PHY-1820867.

# A    The Chern-Simons invariants

The Chern-Simons form is defined by,

$$\mathrm{CS}(\omega) := \mathrm{tr}\left(\omega \wedge d\omega + \frac{2}{3}\omega \wedge \omega \wedge \omega\right), \tag{A.1}$$

where the spin connection is

$$\omega_\mu{}^{ab} := \frac{1}{2}(\Omega_{\mu\nu\rho} - \Omega_{\nu\rho\mu} + \Omega_{\rho\mu\nu})e^{\nu a}e^{\rho b}, \tag{A.2}$$

$$\Omega_{\mu\nu\rho} := \left(\partial_\mu e_\nu{}^a - \partial_\nu e_\mu{}^a\right)e_{\rho a}. \tag{A.3}$$

Here, $e^a$ are a set of vielbein such that

$$e^a = e_\mu{}^a dx^\mu, \tag{A.4}$$

$$g_{\mu\nu} = e_\mu{}^a e_\nu{}^b \delta_{ab}. \tag{A.5}$$

For Lorentzian manifolds, we replace $\delta_{ab}$ in (A.5) with $\eta_{ab}$. The inverse $e_b$ satisfies:

$$e_\mu{}^a e^\mu{}_b = \delta^a_b. \tag{A.6}$$

We will use the torsionful spin connection,

$$\omega_+ := \omega + \frac{1}{2}H_3, \tag{A.7}$$

when computing the Chern-Simons form

$$\mathrm{CS}(\omega_+) = \mathrm{tr}\left(\omega_+ \wedge d\omega_+ + \frac{2}{3}\omega_+ \wedge \omega_+ \wedge \omega_+\right). \tag{A.8}$$

## A.1    Chern-Simons form on $S^3$

The metric on $S^3$ with radius $L$ is

$$ds^2 = g_{\mu\nu}dx^\mu dx^\nu = L^2 d\psi^2 + L^2 \sin^2\psi d\theta^2 + L^2 \sin^2\psi \sin^2\theta d\phi^2. \tag{A.9}$$

We choose vielbeins

$$e^\psi = Ld\psi, \tag{A.10}$$

$$e^\theta = L\sin\psi d\theta, \tag{A.11}$$

$$e^\phi = L\sin\psi \sin\theta d\phi. \tag{A.12}$$

The inverses are given explicitly by

$$e_\psi = \frac{1}{L}\partial_\psi \,, \tag{A.13}$$

$$e_\theta = \frac{1}{L\sin\psi}\partial_\theta \,, \tag{A.14}$$

$$e_\phi = \frac{1}{L\sin\psi\sin\theta}\partial_\phi \,. \tag{A.15}$$

Computing the spin connection, we get

$$\omega = \begin{pmatrix} 0 & -\cos\psi d\theta & -\sin\theta\cos\psi d\phi \\ \cos\psi d\theta & 0 & -\cos\theta d\phi \\ \sin\theta\cos\psi d\phi & \cos\theta d\phi & 0 \end{pmatrix} \,, \tag{A.16}$$

$$= \frac{1}{L} \begin{pmatrix} 0 & -\cot\psi e^\theta & -\cot\psi e^\phi \\ \cot\psi e^\theta & 0 & -\frac{\cot\theta}{\sin\psi}e^\phi \\ \cot\psi e^\phi & \frac{\cot\theta}{\sin\psi}e^\phi & 0 \end{pmatrix} \,. \tag{A.17}$$

Here rows denote the $a$ index and columns denote the $b$ index in $\omega_\mu{}^{ab}$, and the index ordering is $(\psi,\theta,\phi)$. The Lie algebra-valued one-form $\omega = \omega_\mu dx^\mu$.

We now consider the $H_3$-flux through the sphere, given by

$$H_3 = \frac{2\alpha' n_5}{L^3}\epsilon_{S^3} \,, \tag{A.18}$$

where $\epsilon_{S^3} = e^\psi \wedge e^\theta \wedge e^\phi$. Here we can ignore $\alpha'$ corrections to $H_3$ since we are computing the leading correction to (2.3) in $\alpha'$. The components are determined by

$$H_3 = \frac{1}{3!}H_{abc}e^a \wedge e^b \wedge e^c \,. \tag{A.19}$$

This gives the Lie algebra-valued one-form, which is an ingredient in $\omega_+$:

$$\frac{1}{2}H_3 = \frac{\alpha' n_5}{L^3} \begin{pmatrix} 0 & e^\phi & -e^\theta \\ -e^\phi & 0 & e^\psi \\ e^\theta & -e^\psi & 0 \end{pmatrix} \,. \tag{A.20}$$

Here, again rows give the first index $a$ and columns give the second index $b$ in $H_\mu{}^{ab}$. So we have defined $H_3 = H_\mu dx^\mu$ as the Lie algebra-valued one-form.

Let us define the dimensionless constant

$$\xi := \frac{\alpha' n_5}{L^2} = \frac{L}{\mathfrak{L}} \,, \tag{A.21}$$

for convenience. Note that $\xi = \pm 1$ for the tree-level solution, where the sign is given by the sign of $n_5$. For the one-loop solution at small coupling,

$$\xi = \pm \left[ 1 + \frac{1}{4} \lambda |n_5| g_s^2 + \mathcal{O}(\lambda N g_s^2) \right] . \tag{A.22}$$

The torsionful spin connection is

$$\omega_+ = \omega + \frac{1}{2} H_3 = \frac{1}{L} \begin{pmatrix} 0 & \xi e^\phi - \cot \psi e^\theta & -\xi e^\theta - \cot \psi e^\phi \\ -\xi e^\phi + \cot \psi e^\theta & 0 & \xi e^\psi - \frac{\cot \theta}{\sin \psi} e^\phi \\ \xi e^\theta + \cot \psi e^\phi & -\xi e^\psi + \frac{\cot \theta}{\sin \psi} e^\phi & 0 \end{pmatrix} . \tag{A.23}$$

Computing the Chern-Simons form gives

$$\text{CS}(\omega_+) = \text{tr} \left( \omega_+ \wedge d\omega_+ + \frac{2}{3} \omega_+ \wedge \omega_+ \wedge \omega_+ \right) \tag{A.24}$$

$$= 2\xi \sin \theta \left[ 2(\xi^2 - 1) \sin^2 \psi - 1 \right] d\psi \wedge d\theta \wedge d\phi . \tag{A.25}$$

Taking the tree-level case with $|\xi| = 1$ and integrating gives

$$\left| \int_{S^3} \text{CS}(\omega_+) \right| = 8\pi^2 , \tag{A.26}$$

which explicitly shows that the Chern-Simons form is non-trivial in cohomology.

## A.2   Chern-Simons form on $\text{AdS}_3$

The metric on $\text{AdS}_3$ is

$$ds^2 = -L_{\text{AdS}}^2 \cosh^2 \rho \, dt^2 + L_{\text{AdS}}^2 d\rho^2 + L_{\text{AdS}}^2 \sinh^2 \rho \, d\phi^2 . \tag{A.27}$$

We choose vielbein,

$$e^t = L_{\text{AdS}} \cosh \rho \, dt , \tag{A.28}$$

$$e^\rho = L_{\text{AdS}} d\rho , \tag{A.29}$$

$$e^\phi = L_{\text{AdS}} \sinh \rho \, d\phi , \tag{A.30}$$

with inverses,

$$e_t = \frac{1}{L_{\text{AdS}} \cosh \rho} \partial_t , \tag{A.31}$$

$$e_\rho = \frac{1}{L_{\text{AdS}}} \partial_\rho , \tag{A.32}$$

$$e_\phi = \frac{1}{L_{\text{AdS}} \sin \rho} \partial_\phi . \tag{A.33}$$

Computing the spin connection gives,

$$\omega = \begin{pmatrix} 0 & \sinh\rho dt & 0 \\ -\sinh\rho dt & 0 & -\cosh\rho d\phi \\ 0 & \cosh\rho d\phi & 0 \end{pmatrix}, \tag{A.34}$$

$$= \frac{1}{L_{\text{AdS}}} \begin{pmatrix} 0 & \tanh\rho e^t & 0 \\ -\tanh\rho e^t & 0 & -\coth\rho e^\phi \\ 0 & \coth\rho e^\phi & 0 \end{pmatrix}. \tag{A.35}$$

The $H_3$-flux through AdS$_3$ is given by

$$H_3^{\text{electric}} = \frac{8\pi\alpha'^3 g_s^2}{rL^3\hat{L}^3} n_1\,\epsilon_3\,, \tag{A.36}$$

where $\epsilon_3 = e^t \wedge e^\rho \wedge e^\phi$. The corresponding Lie algebra-valued one-form is

$$\frac{1}{2}H_3 = \frac{4\pi\alpha'^3 g_s^2}{rL^3\hat{L}^3} n_1 \begin{pmatrix} 0 & e^\phi & -e^\rho \\ -e^\phi & 0 & e^t \\ e^\rho & -e^t & 0 \end{pmatrix}. \tag{A.37}$$

Let us define the dimensionless constant

$$\xi := \frac{4\pi\alpha'^3 g_s^2 L_{\text{AdS}}}{rL^3\hat{L}^3} n_1 = \frac{L_{\text{AdS}}}{\mathfrak{L}_{\text{AdS}}}\,. \tag{A.38}$$

Note that $\xi = \pm 1$ for the tree-level solution, where the sign is given by the sign of $n_1$. For the one-loop solution at small coupling,

$$\xi = \pm \left[1 + \frac{1}{8}\lambda \left(\frac{L_{\text{AdS}}^2}{\alpha'} - 3\frac{\alpha'}{L_{\text{AdS}}^2}\right) g_s^2 + \mathcal{O}(\lambda N g_s^2)\right]. \tag{A.39}$$

The torsionful spin connection is

$$\omega_+ = \frac{1}{L_{\text{AdS}}} \begin{pmatrix} 0 & \xi e^\phi + \tanh\rho e^t & -\xi e^\rho \\ -\xi e^\phi - \tanh\rho e^t & 0 & \xi e^t - \coth\rho e^\phi \\ \xi e^\rho & -\xi e^t + \coth\rho e^\phi & 0 \end{pmatrix}. \tag{A.40}$$

We compute the Chern-Simons form finding:

$$\text{CS}(\omega_+) = \left[2\xi(1+\xi^2)\sinh 2\rho\right] dt \wedge d\rho \wedge d\phi\,. \tag{A.41}$$

# B  The one-loop potential

## B.1  Modular identites

Let $q := e^{2\pi i \tau}$. We define the *Dedekind eta function* as usual by

$$\eta(\tau) := q^{1/24} \prod_{n=1}^{\infty} (1 - q^n) \,. \tag{B.1}$$

The *Jacobi theta function* is defined by

$$\frac{\vartheta \begin{bmatrix} \alpha \\ \beta \end{bmatrix} (\tau)}{\eta(\tau)} := e^{2\pi i \alpha \beta} q^{\alpha^2/2 - 1/24} \prod_{n=1}^{\infty} (1 + e^{2\pi i \beta} q^{n + \alpha - 1/2}) \times$$
$$(1 + e^{-2\pi i \beta} q^{n - \alpha - 1/2}), \quad |\alpha|, |\beta| \leq 1/2 \,. \tag{B.2}$$

The definition of the Jacobi theta function is extended to $|\alpha|, |\beta| > 1/2$ by using the identity

$$\vartheta \begin{bmatrix} \alpha + m \\ \beta + n \end{bmatrix} (\tau) = e^{2\pi i n \alpha} \vartheta \begin{bmatrix} \alpha \\ \beta \end{bmatrix} (\tau), \qquad m, n \in \mathbb{Z} \,. \tag{B.3}$$

Under the modular transformation $T : \tau \mapsto \tau + 1$, the functions transform as follows:

$$\eta(\tau) \mapsto e^{\pi i /12} \eta(\tau) \,, \tag{B.4}$$

$$\vartheta \begin{bmatrix} \alpha \\ \beta \end{bmatrix} (\tau) \mapsto e^{-\pi i (\alpha^2 - \alpha)} \vartheta \begin{bmatrix} \alpha \\ \beta + \alpha - 1/2 \end{bmatrix} (\tau) \,. \tag{B.5}$$

Under the modular transformation $S : \tau \mapsto -1/\tau$, the functions transform as

$$\eta(\tau) \mapsto \sqrt{-i\tau} \eta(\tau) \,, \tag{B.6}$$

$$\vartheta \begin{bmatrix} \alpha \\ \beta \end{bmatrix} (\tau) \mapsto \sqrt{-i\tau} e^{2\pi i \alpha \beta} \vartheta \begin{bmatrix} -\beta \\ \alpha \end{bmatrix} (\tau) \,. \tag{B.7}$$

Lastly, let $\Lambda$ be a rank $n$ lattice, let $A$ be an $n \times n$ non-singular matrix and $v$ a vector in the $\mathbb{R}$-span of $\Lambda$. The *Poisson resummation* identity states that

$$\sum_{p \in \Lambda} e^{-\pi p^T A p + 2\pi i p \cdot v} = \frac{1}{\text{vol}(\Lambda) \sqrt{\det A}} \sum_{p \in \Lambda^*} e^{-\pi (p+v)^T A^{-1} (p+v)} \,. \tag{B.8}$$

## B.2 Orbifolding review

Orbifolding by a group $G$ is performed by taking the $G$-invariant states of the parent theory and then adding twisted sectors for each conjugacy class. For the original papers, see [44–46].

In our case, $G$ is a cyclic group, so we make the assumption $G \cong \mathbb{Z}_N$ in this short review. We use modular invariance as the guiding principle in our presentation of the orbifolding procedure as in [47]. In this approach, we define traces over the Hilbert space of the parent theory for each $g \in G$

$$Z_1^g := \text{tr}\left(\rho(g) q^{L_0 - c/24} \bar{q}^{\bar{L}_0 - \bar{c}/24}\right), \tag{B.9}$$

where $\rho$ is the representation of $G$ on the Hilbert space of the parent theory. The *untwisted sector* is formed from the $G$-invariant states of the parent theory by inserting the projector $\Pi_G := \frac{1}{|G|} \sum_{g \in G} \rho(g)$ in the trace

$$Z_1 := \text{tr}\left(\Pi_G \, q^{L_0 - c/24} \bar{q}^{\bar{L}_0 - \bar{c}/24}\right) = \frac{1}{|G|} \sum_{g \in G} Z_1^g. \tag{B.10}$$

Even though $Z_1$ is invariant under the modular transformation $T : \tau \mapsto \tau + 1$, it is usually not invariant under $S : \tau \mapsto -1/\tau$. To ensure modular invariance, we define twisted sector partial traces using

$$Z_h^g(\tau + 1) = Z_h^{gh^{-1}}(\tau), \tag{B.11}$$

$$Z_h^g\left(-\frac{1}{\tau}\right) = Z_g^{h^{-1}}(\tau). \tag{B.12}$$

Here, the subscripts may not always be periodic modulo $N$, but they will always be periodic modulo $N^2$. Define the twisted sector partition functions as

$$Z_h := \frac{1}{|G|} \sum_{g \in G} Z_h^g, \tag{B.13}$$

and the total partition function of the orbifolded theory as

$$Z := \sum_{h \in G} Z_h = \frac{1}{|G|} \sum_{h,g \in G} Z_h^g. \tag{B.14}$$

It only remains to check that $Z$ is modular invariant to verify that the orbifolded theory is not anomalous. This would mean checking that $Z_h^g$ for $h, g \in G$ form modular orbits, i.e. that every subscript is periodic modulo $N$.

## B.3 The partition function

We construct the partition function of the non-supersymmetric $O(16) \times O(16)$ heterotic string on $\mathbb{R}^9 \times S^1$. We start with a supersymmetric heterotic string on $S^1$ and then do a $\mathbb{Z}_2$ orbifold as in [23]. The orbifold group $G \cong \mathbb{Z}_2$ has the generator $g := (-1)^F \rho(\delta)$, where $\rho(\delta)$ is the representation of the shift vector $\delta \in \Gamma^{17,1} \cong \Gamma^{16,0} \oplus \Gamma^{1,1}$ acting on the Hilbert space. There are two choices for $\delta$ corresponding to the two $\Gamma^{16}$ lattices:

$$\delta = \begin{cases} (1, 0^7, 1, 0^7, 0; 0) + k - \bar{k} & \Gamma^{16} \cong E_8 \times E_8 \,, \\ ((1/2)^8, 0^8, 0; 0) + k - \bar{k} & \Gamma^{16} \cong D_{16}^+ \,, \end{cases} \tag{B.15}$$

where $k, \bar{k}$ are null vectors with $k \cdot \bar{k} = 1$ that generate $\Gamma^{1,1}$. We choose the $E_8 \times E_8$ heterotic string in our computation.

The $E_8 \times E_8$ heterotic string partition function on $\mathbb{R}^9 \times S^1$ is

$$Z_{S^1} = \eta^{-24} \left( \sum_{(p_L; p_R) \in \Gamma^{17,1}} q^{p_L^2/2} \bar{q}^{p_R^2/2} \right)$$

$$\times \bar{\eta}^{-12} \frac{1}{2} \left( \bar{\vartheta} \begin{bmatrix} 0 \\ 0 \end{bmatrix}^4 - \bar{\vartheta} \begin{bmatrix} 0 \\ 1/2 \end{bmatrix}^4 - \bar{\vartheta} \begin{bmatrix} 1/2 \\ 0 \end{bmatrix}^4 - \bar{\vartheta} \begin{bmatrix} 1/2 \\ 1/2 \end{bmatrix}^4 \right). \tag{B.16}$$

Now we consider the action of $g = (-1)^F \rho(\delta)$. The $(-1)^F$ acts only on the fermionic oscillators, and will only change the signs of the $\bar{\vartheta}$ functions accordingly. The shift vector acts on the winding-momentum ground states as a phase shift by $e^{2\pi i p \cdot \delta}$ where $p \in \Gamma^{17,1}$. We get

$$Z_1^g = \eta^{-24} \left( \sum_{(p_L; p_R) \in \Gamma^{17,1}} e^{2\pi i p \cdot \delta} q^{p_L^2/2} \bar{q}^{p_R^2/2} \right)$$

$$\times \bar{\eta}^{-12} \frac{1}{2} \left( \bar{\vartheta} \begin{bmatrix} 0 \\ 1/2 \end{bmatrix}^4 - \bar{\vartheta} \begin{bmatrix} 0 \\ 0 \end{bmatrix}^4 - \bar{\vartheta} \begin{bmatrix} 1/2 \\ 1/2 \end{bmatrix}^4 - \bar{\vartheta} \begin{bmatrix} 1/2 \\ 0 \end{bmatrix}^4 \right). \tag{B.17}$$

Next we compute $Z_g^1$ using (B.12) and the identities in Appendix B.1,

$$Z_g^1 = \eta^{-24} \left( \sum_{(p_L; p_R) \in \Gamma^{17,1}} q^{(p_L + \delta)^2/2} \bar{q}^{p_R^2/2} \right)$$

$$\times \bar{\eta}^{-12} \frac{1}{2} \left( \bar{\vartheta} \begin{bmatrix} -1/2 \\ 0 \end{bmatrix}^4 - \bar{\vartheta} \begin{bmatrix} 0 \\ 0 \end{bmatrix}^4 - \bar{\vartheta} \begin{bmatrix} -1/2 \\ 1/2 \end{bmatrix}^4 - \bar{\vartheta} \begin{bmatrix} 0 \\ 1/2 \end{bmatrix}^4 \right). \tag{B.18}$$

Note that the spacetime bosonic oscillators produce an extra factor of $|\tau|^{-7}$ under the $S$ transformation because

$$\eta^{-7}\bar{\eta}^{-7} \mapsto \eta(\sqrt{-i\tau})^{-7}\eta^{-7}(\sqrt{i\tau})^{-7}\bar{\eta}^{-7}\,, \tag{B.19}$$

$$= |\tau|^{-7}\eta^{-7}\bar{\eta}^{-7}\,, \tag{B.20}$$

which is canceled out by the factor from the integration measure

$$\frac{d^2\tau}{\tau_2^{11/2}} \mapsto \frac{d^2\tau}{\tau_2^{11/2}}|\tau|^7\,. \tag{B.21}$$

Therefore, we suppress this extra factor of $|\tau|^{-7}$ in the $S$ transformation of $Z_1^g$.

We next do a $T^{-1}$ transformation on $Z_g^1$ to get

$$Z_g^g = \eta^{-24}\left(\sum_{(p_L;p_R)\in\Gamma^{17,1}} e^{2\pi i p_L\cdot\delta}q^{(p_L+\delta)^2/2}\bar{q}^{p_R^2/2}\right)$$
$$\times \bar{\eta}^{-12}\frac{1}{2}\left(\bar{\vartheta}\begin{bmatrix}-1/2\\-1\end{bmatrix}^4 + \bar{\vartheta}\begin{bmatrix}0\\-1/2\end{bmatrix}^4 - \bar{\vartheta}\begin{bmatrix}-1/2\\-1/2\end{bmatrix}^4 + \bar{\vartheta}\begin{bmatrix}0\\0\end{bmatrix}^4\right)\,. \tag{B.22}$$

Here we used

$$(p_L+\delta)^2 - p_R^2 \equiv 2p_L\cdot\delta \pmod 2\,, \tag{B.23}$$

to simplify the lattice summation term.

Finally, the total partition function is

$$Z = \frac{1}{2}(Z_{S_1} + Z_1^g + Z_g^1 + Z_g^g)\,. \tag{B.24}$$

We can check that the orbifolded theory is non-anomalous by computing the modular orbit of each partial trace by using the identities of Appendix B.1. The orbit structures are given in the diagram below.

$$T,S \circlearrowright Z_{S^1} \qquad T \circlearrowright Z_1^g \xleftarrow{S} Z_g^1 \xleftarrow{T} Z_g^g \circlearrowright S$$

## B.4   The one-loop potential

The one-loop contribution to the cosmological constant is computed as the integral of the partition function over the fundamental domain with the appropriate measure [48, 49] as

$$\Lambda_d = \frac{1}{2(2\pi\sqrt{\alpha'})^d}\int_{\mathcal{F}}\frac{d^2\tau}{\tau_2^{d/2+1}}Z\,. \tag{B.25}$$

In our case, $d = 9$. Evaluating $\Lambda_9$ at the minimum of the potential with $r = \sqrt{\alpha'}$ gives,

$$\Lambda_9 = \frac{1}{2(2\pi\sqrt{\alpha'})^9} \int_{\mathcal{F}} \frac{d^2\tau}{\tau_2^{11/2}} Z = (2.29 \times 10^{-5}) \, \alpha'^{-9/2} \,. \tag{B.26}$$

Figure 5 shows that $r = \sqrt{\alpha'}$ really is a local minimum of $\Lambda_9$. The cosmological constant

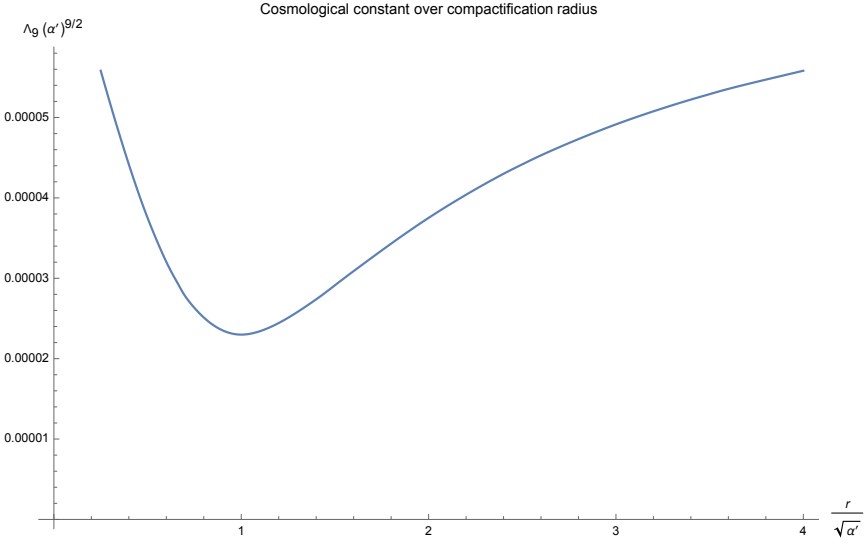

Figure 5: Value of $\Lambda_9$ versus compactification radius $r$ in string units. We see that the self-dual radius is the minimum. Also notice the T-duality in the graph given by the symmetry $r/\alpha'^{1/2} \leftrightarrow \alpha'^{1/2}/r$.

shows up in the 9-dimensional action as

$$-\int \sqrt{-g}\Lambda_9 = -\frac{1}{2\kappa_9^2} \left( \kappa_9^2 \int 2\sqrt{-g}\,\Lambda_9 \right) \,, \tag{B.27}$$

$$\kappa_9^2 = \frac{\frac{1}{2}(2\pi)^7 \alpha'^4 e^{2\phi_0}}{2\pi r} \,, \tag{B.28}$$

$$r = \sqrt{\alpha} \,. \tag{B.29}$$

From now on, we consider the cosmological constant with dimensions $[L]^{-2}$ obtained by factoring out $\kappa_9^2$,

$$\Lambda := \kappa_9^2 \Lambda_9 = \lambda \frac{g_s^2}{\alpha'}, \qquad \lambda \approx 0.705 \,. \tag{B.30}$$

Here, $\lambda$ is a dimensionless constant.

Integrating over $S^3 \times \hat{S}^3$, we get

$$-\frac{1}{2\kappa_9^2} \int d^9x \, 2\sqrt{-g}\Lambda = -\frac{1}{2\kappa_9^2} (2\pi^2 L^3)(2\pi^2 \hat{L}^3) \int d^3x \, 2\sqrt{-g}\, e^{3\chi+3\hat{\chi}}\Lambda \,. \tag{B.31}$$

After a redefinition of the gravitational coupling, we find

$$= -\frac{1}{2\kappa_3^2} \int d^3x \; 2\sqrt{-g}\Lambda_3 \,, \tag{B.32}$$

where

$$\kappa_3^2 = \frac{\kappa_9^2}{(2\pi^2 L^3)(2\pi^2 \hat{L}^3)} \,. \tag{B.33}$$

Also notice that

$$\Lambda_3 = \Lambda \,. \tag{B.34}$$

The analysis so far was in string-frame. Rescaling the metric $g = \text{vol}^{-2}\hat{g}$ gives the following term in the Einstein-frame action

$$-\frac{1}{2\kappa_3^2} \int d^3x \; 2\sqrt{-g}e^{3\chi+3\hat{\chi}}\Lambda_3 = -\frac{1}{2\kappa_3^2} \int d^3x \; 2\sqrt{-\hat{g}} \,\text{vol}^{-3}\, e^{3\chi+3\hat{\chi}}\Lambda_3 \,. \tag{B.35}$$

In particular, the one-loop potential is

$$V^{1-\text{loop}}(\phi, \sigma, \chi, \hat{\chi}) := \text{vol}^{-3}\, e^{3\chi+3\hat{\chi}} \times 2\Lambda = \text{vol}^{-3}\, e^{3\chi+3\hat{\chi}} \times 2\lambda\frac{g_s^2}{\alpha'} \,. \tag{B.36}$$

# C   Coupled scalar sector analysis

This Appendix is identical to [38, Appendix B] except for the modifications we now describe.

First, we have length scales associated to fluxes, which can be written using the Einstein equations (5.9) as

$$\mathfrak{L}_{\text{AdS,o}}^{-1} = \sqrt{L_{\text{AdS,o}}^{-2} - \frac{1}{2}\Lambda} \,, \tag{C.1}$$

$$\mathfrak{L}_{\text{o}}^{-1} = \sqrt{L_{\text{o}}^{-2} + \frac{1}{2}\Lambda} \,, \tag{C.2}$$

$$\hat{\mathfrak{L}}_{\text{o}}^{-1} = \sqrt{\hat{L}_{\text{o}}^{-2} + \frac{1}{2}\Lambda} \,. \tag{C.3}$$

The equations in [38, Appendix B] are modified such that their sphere and AdS length scales appear here as either $L_{\text{AdS,o}}, L_{\text{o}}, \hat{L}_{\text{o}}$ or $\mathfrak{L}_{\text{AdS,o}}, \mathfrak{L}_{\text{o}}, \hat{\mathfrak{L}}_{\text{o}}$. Furthermore, we find additional terms proportional to $\Lambda$.

We should also note that even though (C.1) seems as though it could be imaginary, $\Lambda = \lambda g_{s,\text{o}}^2/\alpha'$ can not get arbitrarily large due to the bound on $g_{s,\text{o}}^2$, so $\mathfrak{L}_{\text{AdS,o}}$ is always real.

## C.1 Spherical harmonics

Spherical harmonics have the following properties:

$$\Box_x Y^{(\ell\,0)} = (1 - (\ell+1)^2)Y^{(\ell\,0)} \,, \tag{C.4}$$

$$\Box_x Y_a^{(\ell\,\pm 1)} = (2 - (\ell+1)^2)Y_a^{(\ell\,\pm 1)} \,, \tag{C.5}$$

$$\nabla^a Y_a^{(\ell\,\pm 1)} = 0 \,, \tag{C.6}$$

$$\Box_x Y_{ab}^{(\ell\,\pm 2)} = (3 - (\ell+1)^2)Y_{ab}^{(\ell\,\pm 2)} \,, \tag{C.7}$$

$$\nabla^a Y_{ab}^{(\ell\,\pm 2)} = 0 \,, \tag{C.8}$$

$$g^{ab} Y_{ab}^{(\ell\,\pm 2)} = 0 \,, \tag{C.9}$$

$$\epsilon_a{}^{bc}\partial_b Y_c^{(\ell\,\pm 1)} = \pm(\ell+1)Y_a^{(\ell\,\pm 1)} \,. \tag{C.10}$$

Here, $\Box_x$ is the Laplace operator on $S^3$. The Laplace operator on $\hat{S}^3$ is denoted as $\Box_{\hat{x}}$ and on $\mathrm{AdS}_3$ by $\Box_0$.

## C.2 Equations of motion

**Dilaton:**

$$
\begin{aligned}
0 = &-(\Box_0 + \Box_x + \Box_{\hat{x}})\phi + \frac{1}{4}\Box_0(2M + 3N + 3P) \\
&+ \frac{1}{4}\Box_x(3M + 2N + 3P) + \frac{1}{4}\Box_{\hat{x}}(3M + 3N + 2P) \\
&- \frac{1}{2}\mathfrak{L}_{\mathrm{AdS,o}}^{-1}\nabla_\lambda U^\lambda + \frac{1}{2}\mathfrak{L}_{\mathrm{o}}^{-1}\nabla_a V^a + \frac{1}{2}\hat{\mathfrak{L}}_{\mathrm{o}}^{-1}\nabla_i W^i \\
&- \frac{1}{4}\nabla_\mu\nabla_\nu H^{\mu\nu} - \frac{1}{4}\nabla_i\nabla_j L^{ij} - \frac{1}{2}\nabla_\mu\nabla_i S^{\mu i} - \frac{3}{4}\Lambda(M + N + P) \,.
\end{aligned} \tag{C.11}
$$

**Metric:**

$\mu\nu$-trace component:

$$
\begin{aligned}
0 = &- \left(\Box_0 + 3\Box_x + 3\Box_{\hat{x}} - 12 L_{\mathrm{AdS,o}}^{-2}\right)M - \Box_0(3N + 3P - 4\phi) - \Box_x\left(3N + \frac{9}{2}P - 6\phi\right) \\
&- \Box_{\hat{x}}\left(\frac{9}{2}N + 3P - 6\phi\right) - 3\mathfrak{L}_{\mathrm{AdS,o}}^{-1}\nabla_\mu U^\mu - 3\mathfrak{L}_{\mathrm{o}}^{-1}\nabla_a V^a - 3\hat{\mathfrak{L}}_{\mathrm{o}}^{-1}\nabla_i W^i \\
&+ \frac{1}{2}\nabla_\mu\nabla_\nu H^{\mu\nu} + \frac{3}{2}\nabla_i\nabla_j L^{ij} + 2\nabla_\mu\nabla_i S^{\mu i} + \frac{9}{2}\Lambda(-M + N + P) - 6\Lambda\phi \,.
\end{aligned} \tag{C.12}
$$

$\mu\nu$-traceless component:

$$
\begin{aligned}
0 = &-(\Box_0 + \Box_x + \Box_{\hat{x}} - 4L_{\mathrm{AdS,o}}^{-2})H_{\mu\nu} + 2\nabla_\rho\nabla_{\{\mu}H_{\nu\}}{}^\rho + 2\nabla_i\nabla_{\{\mu}S_{\nu\}}{}^i \\
&- \nabla_{\{\mu}\nabla_{\nu\}}(M + 3N + 3P - 4\phi) \,.
\end{aligned} \tag{C.13}
$$

$ab$-trace component:

$$0 = -\left(\Box_0 + \frac{1}{3}\Box_x + \Box_{\hat{x}} + 4L_\circ^{-2}\right)N - \Box_0\left(M + \frac{3}{2}P - 2\phi\right) - \Box_x\left(M + P - \frac{4}{3}\phi\right)$$

$$- \Box_{\hat{x}}\left(\frac{3}{2}M + P - 2\phi\right) + \mathfrak{L}_{\text{AdS},\circ}^{-1}\nabla_\lambda U^\lambda + \mathfrak{L}_\circ^{-1}\nabla_a V^a - \hat{\mathfrak{L}}_\circ^{-1}\nabla_i W^i$$

$$+ \frac{1}{2}\nabla_\mu\nabla_\nu H^{\mu\nu} + \frac{1}{2}\nabla_i\nabla_j L^{ij} + \nabla_\mu\nabla_i S^{\mu i} + \frac{3}{2}\Lambda(M - N + P) - 2\Lambda\phi \ . \tag{C.14}$$

$ab$-traceless component:

$$0 = -(\Box_0 + \Box_x + \Box_{\hat{x}} - 2L_\circ^{-2})K_{ab} - \nabla_{\{a}\nabla_{b\}}(3M + N + 3P - 4\phi)$$

$$+ 2\nabla_\mu\nabla_{\{a}R^\mu{}_{b\}} + 2\nabla_i\nabla_{\{a}T_{b\}}{}^i \ . \tag{C.15}$$

$ij$-trace component:

$$0 = -\left(\Box_0 + \Box_x + \frac{1}{3}\Box_{\hat{x}} + 4\hat{L}_\circ^{-2}\right)P - \Box_0\left(M + \frac{3}{2}N - 2\phi\right) - \Box_x\left(\frac{3}{2}M + N - 2\phi\right)$$

$$- \Box_{\hat{x}}\left(M + N - \frac{4}{3}\phi\right) + \mathfrak{L}_{\text{AdS},\circ}^{-1}\nabla_\lambda U^\lambda - \mathfrak{L}_\circ^{-1}\nabla_a V^a + \hat{\mathfrak{L}}_\circ^{-1}\nabla_i W^i$$

$$+ \frac{1}{2}\nabla_\mu\nabla_\nu H^{\mu\nu} + \frac{1}{6}\nabla_i\nabla_j L^{ij} + \frac{2}{3}\nabla_\mu\nabla_i S^{\mu i} + \frac{3}{2}\Lambda(M + N - P) - 2\Lambda\phi \ . \tag{C.16}$$

$ij$-traceless component:

$$0 = -(\Box_0 + \Box_x + \Box_{\hat{x}} + 4\hat{L}_\circ^{-2})L_{ij} - \nabla_{\{i}\nabla_{j\}}(3M + 3N + P - 4\phi)$$

$$+ 2\nabla_k\nabla_{\{i}L_{j\}}{}^k + 2\nabla_\mu\nabla_{\{i}S^\mu{}_{j\}} \ . \tag{C.17}$$

$\mu a$-component:

$$0 = -\left(\Box_0 + \Box_x + \Box_{\hat{x}} - 2L_\circ^{-2} - 2\Lambda\right)R_{\mu a}$$

$$+ \nabla_\lambda\nabla_\mu R^\lambda{}_a + \nabla_\lambda\nabla_a H_\mu{}^\lambda + \nabla_i\nabla_\mu T_a{}^i + \nabla_i\nabla_a S_\mu{}^i$$

$$- \nabla_\mu\nabla_a(2M + 2N + 3P - 4\phi) + 2\mathfrak{L}_{\text{AdS},\circ}^{-1}\nabla_a U_\mu - 2\mathfrak{L}_\circ^{-1}\nabla_\mu V_a$$

$$- 2\mathfrak{L}_\circ^{-1}\epsilon_{abc}\nabla^c C_\mu{}^b + 2\mathfrak{L}_{\text{AdS},\circ}^{-1}\epsilon_{\mu\lambda\nu}\nabla^\nu C^\lambda{}_a \ . \tag{C.18}$$

$\mu i$-component:

$$0 = -\left(\Box_0 + \Box_x + \Box_{\hat{x}} - 2\Lambda\right)S_{\mu i}$$

$$+ \nabla_\lambda\nabla_\mu S^\lambda{}_i + \nabla_j\nabla_\mu L_i{}^j + \nabla_\lambda\nabla_i H_\mu{}^\lambda + \nabla_j\nabla_i S_\mu{}^j$$

$$- \nabla_\mu\nabla_i(2M + 3N + 2P - 4\phi) + 2\mathfrak{L}_{\text{AdS},\circ}^{-1}\nabla_i U_\mu - 2\hat{\mathfrak{L}}_\circ^{-1}\nabla_\mu W_i$$

$$- 2\hat{\mathfrak{L}}_\circ^{-1}\epsilon_{ijk}\nabla^k D_\mu{}^j + 2\mathfrak{L}_{\text{AdS},\circ}^{-1}\epsilon_{\mu\lambda\nu}\nabla^\nu D^\lambda{}_i \ . \tag{C.19}$$

$ai$-component:

$$0 = - \left(\Box_0 + \Box_x + \Box_{\hat{x}} - 2L_\circ^{-2} - 2\Lambda\right) T_{ai}$$
$$+ \nabla_j \nabla_a L_i{}^j + \nabla_\mu \nabla_i R^\mu{}_a + \nabla_\mu \nabla_a S^\mu{}_i + \nabla_j \nabla_i T_a{}^j$$
$$- \nabla_i \nabla_a \left(3M + 2N + 2P - 4\phi\right) - 2\hat{\mathfrak{L}}_\circ^{-1} \nabla_a W_i - 2\mathfrak{L}_\circ^{-1} \nabla_i V_a$$
$$+ 2\mathfrak{L}_\circ^{-1} \epsilon_{abc} \nabla^c E^b{}_i - 2\hat{\mathfrak{L}}_\circ^{-1} \epsilon_{ijk} \nabla^k E_a{}^j \ . \tag{C.20}$$

**Kalb-Ramond field:**

$\mu\nu$-component (contracted with $\epsilon^{\mu\nu\lambda}$):

$$0 = -\nabla^\lambda \nabla_\mu U^\mu - (\Box_x + \Box_{\hat{x}}) U^\lambda + \mathfrak{L}_{\text{AdS},\circ}^{-1} \nabla^\lambda (3M - 3N - 3P + 4\phi)$$
$$+ 2\mathfrak{L}_{\text{AdS},\circ}^{-1} \nabla_i S^{\lambda i} - \epsilon^\lambda{}_{\mu\nu} \nabla_i \nabla^\nu D^{\mu i} \ . \tag{C.21}$$

$ab$-component (contracted with $\epsilon^{abc}$):

$$0 = -\nabla^c \nabla_a V^a - (\Box_0 + \Box_{\hat{x}}) V^c + \mathfrak{L}_\circ^{-1} \nabla^c (-3M + 3N - 3P + 4\phi) + 2\mathfrak{L}_\circ^{-1} \nabla_i T^{ci}$$
$$+ 2\mathfrak{L}_\circ^{-1} \nabla_\mu R^{\mu c} + \epsilon^c{}_{ab} \nabla_i \nabla^b E^{ai} - \epsilon^c{}_{ab} \nabla_\lambda \nabla^b C^{\lambda a} \ . \tag{C.22}$$

$ij$-component (contracted with $\epsilon^{ijk}$):

$$0 = -\nabla^k \nabla_i W^i - (\Box_0 + \Box_x) W^k + \hat{\mathfrak{L}}_\circ^{-1} \nabla^k (-3M - 3N + 3P + 4\phi) + 2\hat{\mathfrak{L}}_\circ^{-1} \nabla_\mu S^{\mu k}$$
$$- \epsilon^k{}_{ij} \nabla_\mu \nabla^j D^{\mu i} \ . \tag{C.23}$$

$\mu a$-component:

$$0 = - \left(\Box_0 + \Box_x + \Box_{\hat{x}} - 2L_\circ^{-2}\right) C_{\mu a} + \nabla_\lambda \nabla_\mu C^\lambda{}_a + \nabla_i \nabla_a D_\mu{}^i - \nabla_\mu \nabla_i E_a{}^i$$
$$+ 2\mathfrak{L}_{\text{AdS},\circ}^{-1} \epsilon_{\mu\lambda\nu} \nabla^\nu R^\lambda{}_a - 2\mathfrak{L}_\circ^{-1} \epsilon_{abc} \nabla^c R_\mu{}^b + \epsilon_{abc} \nabla^c \nabla_\mu V^b - \epsilon_{\mu\lambda\nu} \nabla^\nu \nabla_a U^\lambda \ . \tag{C.24}$$

$\mu i$-component:

$$0 = - \left(\Box_0 + \Box_x + \Box_{\hat{x}}\right) D_{\mu i} + \nabla_\lambda \nabla_\mu D^\lambda{}_i + \nabla_j \nabla_i D_\mu{}^j$$
$$+ 2\mathfrak{L}_{\text{AdS},\circ}^{-1} \epsilon_{\mu\lambda\nu} \nabla^\nu S^\lambda{}_i - 2\hat{\mathfrak{L}}_\circ^{-1} \epsilon_{ijk} \nabla^k S_\mu{}^j + \epsilon_{ijk} \nabla^k \nabla_\mu W^j - \epsilon_{\mu\lambda\nu} \nabla^\nu \nabla_i U^\lambda \ . \tag{C.25}$$

$ai$-component:

$$0 = - \left(\Box_0 + \Box_x + \Box_{\hat{x}} - 2L_\circ^{-2}\right) E_{ai} + \nabla_j \nabla_i E_a{}^j - \nabla_\lambda \nabla_i C^\lambda{}_a + \nabla_\lambda \nabla_a D^\lambda{}_i$$
$$+ 2\mathfrak{L}_\circ^{-1} \epsilon_{abc} \nabla^c T^b{}_i - 2\hat{\mathfrak{L}}_\circ^{-1} \epsilon_{ijk} \nabla^k T_a{}^j - \epsilon_{abc} \nabla^c \nabla_i V^b + \epsilon_{ijk} \nabla^k \nabla_a W^j \ . \tag{C.26}$$

Note that we have broken the symmetry $S^3 \leftrightarrow \hat{S}^3$ by our gauge choice.

## C.3 Scalar parts of the equations

From now on, we discuss the generic case where $\ell \geq 2$ and $\hat{\ell} \geq 2$. We now extract the scalar part of these equations. We have the following scalars:

$$M, \ N, \ P, \ \phi, \ \nabla^\mu U_\mu, \ \nabla^a V_a, \ \nabla^i W_i, \ \nabla_\mu \nabla_\nu H^{\mu\nu}, \ \nabla_i \nabla_j L^{ij}, \ \nabla_\mu \nabla_i S^{\mu i}, \ \nabla_\mu \nabla_i D^{\mu i} \ . \quad \text{(C.27)}$$

We have the decompositions

$$\nabla^a V_a = -L_\circ^{-2} \sum_{\ell,\hat{\ell}} \ell(\ell+2) V^{(\ell 0)(\hat{\ell} 0)} Y_+^{(\ell 0)} Y_-^{(\hat{\ell} 0)} \ , \quad \text{(C.28)}$$

$$\nabla^i W_i = -\hat{L}_\circ^{-2} \sum_{\ell,\hat{\ell}} \hat{\ell}(\hat{\ell}+2) W^{(\ell 0)(\hat{\ell} 0)} Y_+^{(\ell 0)} Y_-^{(\hat{\ell} 0)} \ , \quad \text{(C.29)}$$

$$\nabla^i \nabla^j L_{ij} = \frac{2}{3} \hat{L}_\circ^{-4} \sum_{\ell,\hat{\ell}} (\hat{\ell}-1)\hat{\ell}(\hat{\ell}+2)(\hat{\ell}+3) L^{(\ell 0)(\hat{\ell} 0)} Y_+^{(\ell 0)} Y_-^{(\hat{\ell} 0)} \ . \quad \text{(C.30)}$$

We redefine some variables to clean up notation as

$$\Phi = \phi^{(\ell 0)(\hat{\ell} 0)} \ , \quad \text{(C.31)}$$

$$\mathcal{M} = M^{(\ell 0)(\hat{\ell} 0)} \ , \quad \text{(C.32)}$$

$$\mathcal{N} = N^{(\ell 0)(\hat{\ell} 0)} \ , \quad \text{(C.33)}$$

$$\mathcal{P} = P^{(\ell 0)(\hat{\ell} 0)} \ , \quad \text{(C.34)}$$

$$\mathcal{U} = \mathfrak{L}_{\text{AdS},\circ}^{-1} \nabla^\mu U_\mu^{(\ell 0)(\hat{\ell} 0)} \ , \quad \text{(C.35)}$$

$$\mathcal{H} = \nabla^\mu \nabla^\nu H_{\mu\nu}^{(\ell 0)(\hat{\ell} 0)} \ , \quad \text{(C.36)}$$

$$\mathcal{V} = -\mathfrak{L}_\circ^{-1} L_\circ^{-2} \ell(\ell+2) V^{(\ell 0)(\hat{\ell} 0)} \ , \quad \text{(C.37)}$$

$$\mathcal{W} = -\hat{\mathfrak{L}}_\circ^{-1} \hat{L}_\circ^{-2} \hat{\ell}(\hat{\ell}+2) W^{(\ell 0)(\hat{\ell} 0)} \ , \quad \text{(C.38)}$$

$$\mathcal{S} = -\hat{L}_\circ^{-2} \hat{\ell}(\hat{\ell}+2) \nabla^\mu S_\mu^{(\ell 0)(\hat{\ell} 0)} \ , \quad \text{(C.39)}$$

$$\mathcal{D} = -\hat{L}_\circ^{-2} \hat{\ell}(\hat{\ell}+2) \nabla^\mu D_\mu^{(\ell 0)(\hat{\ell} 0)} \ , \quad \text{(C.40)}$$

$$\mathcal{L} = \frac{2}{3} \hat{L}_\circ^{-4} (\hat{\ell}-1)\hat{\ell}(\hat{\ell}+2)(\hat{\ell}+3) L^{(\ell 0)(\hat{\ell} 0)} \ . \quad \text{(C.41)}$$

There is no ambiguity in this notation since terms with different $\ell$ or $\hat{\ell}$ do not mix. Application of $\nabla^\mu \nabla^a$ to (C.24) gives $\mathcal{D} = 0$, and this then implies the scalar parts of (C.25) and (C.26). Applying $\nabla^a \nabla^b$ on (C.15), we get an algebraic equation relating $\Phi$, $\mathcal{M}$, $\mathcal{N}$ and $\mathcal{P}$. Extracting the scalar part of (C.20) by applying $\nabla^a \nabla^i$ will yield a further algebraic equation. A last algebraic equation will come from a combination of (C.12) and (C.14). We then have four algebraic equations, cutting down the number of scalar fields to seven.

We apply $\nabla^a \nabla^b$ to (C.15) and find that

$$3\mathcal{M} + \mathcal{N} + 3\mathcal{P} - 4\Phi = 0 \ . \quad \text{(C.42)}$$

So we will eliminate the field $\Phi$ in terms of $\mathcal{M}, \mathcal{N}, \mathcal{P}$. The remaining equations written with these replacements are:

**Dilaton:**

$$0 = \Box_0\left(-\frac{1}{4}\mathcal{M} + \frac{1}{2}\mathcal{N}\right) - \frac{1}{4}\left(L_\circ^{-2}\ell(\ell+2) + 2\hat{L}_\circ^{-2}\hat{\ell}(\hat{\ell}+2) + 3\Lambda\right)\mathcal{N}$$
$$+ \frac{1}{4}(\hat{L}_\circ^{-2}\hat{\ell}(\hat{\ell}+2) - 3\Lambda)\mathcal{P} - \frac{3}{4}\Lambda\mathcal{M} - \frac{1}{2}\mathcal{U} + \frac{1}{2}\mathcal{V} + \frac{1}{2}\mathcal{W} - \frac{1}{4}\mathcal{H} - \frac{1}{2}\mathcal{S} - \frac{1}{4}\mathcal{L} . \quad \text{(C.43)}$$

**Metric:**

$\mu\nu$-trace component:

$$0 = \Box_0(2\mathcal{M} - 2\mathcal{N}) + \hat{L}_\circ^{-2}\hat{\ell}(\hat{\ell}+2)\left(-\frac{3}{2}\mathcal{M} + 3\mathcal{N} - \frac{3}{2}\mathcal{P}\right) + 12L_{\text{AdS},\circ}^{-2}\mathcal{M}$$
$$+ L_\circ^{-2}\ell(\ell+2)\left(-\frac{3}{2}\mathcal{M} + \frac{3}{2}\mathcal{N}\right) - 3\mathcal{U} - 3\mathcal{V} - 3\mathcal{W} + \frac{1}{2}\mathcal{H} + 2\mathcal{S} + \frac{3}{2}\mathcal{L}$$
$$+ \Lambda(-9\mathcal{M} + 3\mathcal{N}) . \quad \text{(C.44)}$$

$\mu\nu$-traceless component:

$$0 = \frac{4}{3}\Box_0^2(\mathcal{M} - \mathcal{N}) + \Box_0\left(-4L_{\text{AdS},\circ}^{-2}\mathcal{M} + 4L_{\text{AdS},\circ}^{-2}\mathcal{N} + \frac{1}{3}\mathcal{H} + \frac{4}{3}\mathcal{S}\right)$$
$$+ \left(L_\circ^{-2}\ell(\ell+2) + \hat{L}_\circ^{-2}\hat{\ell}(\hat{\ell}+2)\right)\mathcal{H} - 4L_{\text{AdS},\circ}^{-2}\mathcal{S} . \quad \text{(C.45)}$$

$ab$-trace component:

$$0 = \frac{1}{2}\Box_0(\mathcal{M} - \mathcal{N}) + \frac{1}{2}\hat{L}_\circ^{-2}\hat{\ell}(\hat{\ell}+2)(\mathcal{N} - \mathcal{P}) - 4L_\circ^{-2}\mathcal{N} + \mathcal{U} + \mathcal{V} - \mathcal{W}$$
$$+ \frac{1}{2}\mathcal{H} + \mathcal{S} + \frac{1}{2}\mathcal{L} - 2\Lambda\mathcal{N} . \quad \text{(C.46)}$$

$ij$-trace component:

$$0 = \Box_0\left(\frac{1}{2}\mathcal{M} - \mathcal{N} + \frac{1}{2}\mathcal{P}\right) + \left(\frac{1}{2}L_\circ^{-2}\ell(\ell+2) + \frac{2}{3}\hat{L}_\circ^{-2}\hat{\ell}(\hat{\ell}+2)\right)(\mathcal{N} - \mathcal{P})$$
$$- 4\hat{L}_\circ^{-2}\mathcal{P} + \mathcal{U} - \mathcal{V} + \mathcal{W} + \frac{1}{2}\mathcal{H} + \frac{2}{3}\mathcal{S} + \frac{1}{6}\mathcal{L} + \Lambda\mathcal{N} - 3\Lambda\mathcal{P} . \quad \text{(C.47)}$$

$ij$-traceless component:

$$0 = -\Box_0\mathcal{L} + \frac{4}{3}\hat{L}_\circ^{-4}(\hat{\ell}-1)\hat{\ell}(\hat{\ell}+2)(\hat{\ell}+3)(-\mathcal{N} + \mathcal{P})$$
$$+ \left(L_\circ^{-2}\ell(\ell+2) - \frac{1}{3}\hat{L}_\circ^{-2}\hat{\ell}(\hat{\ell}+2)\right)\mathcal{L} - \frac{4}{3}\hat{L}_\circ^{-2}\left(\hat{\ell}(\hat{\ell}+2) - 3\right)\mathcal{S} . \quad \text{(C.48)}$$

$\mu a$-component:

$$0 = \Box_0 \left( L_\circ^{-2}\ell(\ell+2)(-\mathcal{M}+\mathcal{N}) - 2\mathcal{V} \right) + L_\circ^{-2}\ell(\ell+2)(-2\mathcal{U}-\mathcal{H}-\mathcal{S}) \,. \tag{C.49}$$

$\mu i$-component:

$$\begin{aligned}
0 =&\Box_0 \left( \hat{L}_\circ^{-2}\hat{\ell}(\hat{\ell}+2)(-\mathcal{M}+2\mathcal{N}-\mathcal{P}) - 2\mathcal{W}+\mathcal{L} \right) - \hat{L}_\circ^{-2}\hat{\ell}(\hat{\ell}+2)(\mathcal{H}+2\mathcal{U}) \\
&+ \left( L_\circ^{-2}\ell(\ell+2) + 2\Lambda \right)\mathcal{S} \,.
\end{aligned} \tag{C.50}$$

$ai$-component:

$$\begin{aligned}
0 =& \hat{L}_\circ^{-2}L_\circ^{-2}\ell(\ell+2)\hat{\ell}(\hat{\ell}+2)(-\mathcal{N}+\mathcal{P}) + 2\hat{L}_\circ^{-2}\hat{\ell}(\hat{\ell}+2)\mathcal{V} \\
&+ L_\circ^{-2}\ell(\ell+2)(2\mathcal{W}-\mathcal{S}-\mathcal{L}) \,.
\end{aligned} \tag{C.51}$$

**Kalb-Ramond field:**

$\mu\nu$-component:

$$\begin{aligned}
0 =&\Box_0 \left( 6\mathfrak{L}_{\text{AdS},\circ}^{-2}\mathcal{M} - 2\mathfrak{L}_{\text{AdS},\circ}^{-2}\mathcal{N} - \mathcal{U} \right) + \left( L_\circ^{-2}\ell(\ell+2) + \hat{L}_\circ^{-2}\hat{\ell}(\hat{\ell}+2) \right)\mathcal{U} \\
&+ 2\mathfrak{L}_{\text{AdS},\circ}^{-2}\mathcal{S} \,.
\end{aligned} \tag{C.52}$$

$ab$-component:

$$0 = -\Box_0\mathcal{V} - 4\mathfrak{L}_\circ^{-2}L_\circ^{-2}\ell(\ell+2)\mathcal{N} + \left( L_\circ^{-2}\ell(\ell+2) + \hat{L}_\circ^{-2}\hat{\ell}(\hat{\ell}+2) \right)\mathcal{V} \,. \tag{C.53}$$

$ij$-component:

$$\begin{aligned}
0 =& -\Box_0\mathcal{W} + \hat{\mathfrak{L}}_\circ^{-2}\hat{L}_\circ^{-2}\hat{\ell}(\hat{\ell}+2)\left(2\mathcal{N}-6\mathcal{P}\right) + \left( L_\circ^{-2}\ell(\ell+2) + \hat{L}_\circ^{-2}\hat{\ell}(\hat{\ell}+2) \right)\mathcal{W} \\
&+ 2\hat{\mathfrak{L}}_\circ^{-2}\mathcal{S} \,.
\end{aligned} \tag{C.54}$$

From these equations, (C.44)$-4\times$(C.46) and (C.51) are algebraic and hence impose algebraic relationships among the fields. We use these relations to eliminate the fields $\mathcal{S}$ and $\mathcal{L}$ from the equations. We have

$$0 = \frac{1}{2}\ell(\ell+2) \times \text{(C.44)} - 2\frac{L_\circ}{\mathfrak{L}_\circ} \times \text{(C.53)} + L_\circ^2 \times \text{(C.49)} \,, \tag{C.55}$$

$$0 = 2\hat{\ell}(\hat{\ell}+2) \times \text{(C.47)} + \hat{L}_\circ^2 \times \text{(C.48)} + \hat{L}_\circ^2 \times \text{(C.50)} - 2\frac{\hat{L}_\circ}{\hat{\mathfrak{L}}_\circ} \times \text{(C.54)} \,. \tag{C.56}$$

Thus, we do not have to consider the equations (C.49) and (C.54) any longer. We will not use the equation (C.45), since it contains Laplacian squares.

## C.4 Mass matrix

We are left with seven equations for the seven scalars $\mathcal{M}$, $\mathcal{N}$, $\mathcal{P}$, $\mathcal{U}$, $\mathcal{V}$, $\mathcal{W}$, $\mathcal{H}$, given by (C.43), (C.44), (C.47), (C.50), (C.52), (C.53) and (C.54). They provide the following mass matrix:

$$
\Box_0 \mathcal{M} = \left( L_\circ^{-2}\ell(\ell+2) + \hat{L}_\circ^{-2}\hat{\ell}(\hat{\ell}+2) - 8L_{\text{AdS},\circ}^{-2} + 9\Lambda \right)\mathcal{M} + 3\Lambda\mathcal{P} + \frac{1}{3}\left( 11\Lambda + 16L_\circ^{-2} \right)\mathcal{N}
$$
$$
+ \frac{8}{3}\mathcal{U} - \frac{4}{3}\left( 1 + \frac{\hat{L}_\circ^{-2}\hat{\ell}(\hat{\ell}+2)}{L_\circ^{-2}\ell(\ell+2)} \right)\mathcal{V} \,,
\tag{C.57}
$$

$$
\Box_0 \mathcal{N} = \left( \hat{L}_\circ^{-2}\hat{\ell}(\hat{\ell}+2) + L_\circ^{-2}(\ell(\ell+2)+8) + 7\Lambda \right)\mathcal{N} + 3\Lambda\mathcal{P} - 4\mathcal{V} + 3\Lambda\mathcal{M} \,,
\tag{C.58}
$$

$$
\Box_0 \mathcal{P} = \left( \hat{L}_\circ^{-2}(\hat{\ell}(\hat{\ell}+2)+8) + L_\circ^{-2}\ell(\ell+2) + 9\Lambda \right)\mathcal{P} + \frac{4\hat{\ell}(\hat{\ell}+2)\hat{L}_\circ^{-2}}{3\ell(\ell+2)L_\circ^{-2}}\mathcal{V}
$$
$$
+ \Lambda\mathcal{N} - \frac{8\mathcal{W}}{3} + 3\Lambda\mathcal{M} \,,
\tag{C.59}
$$

$$
\Box_0 \mathcal{U} = -2\mathfrak{L}_{\text{AdS},\circ}^{-2}\mathcal{H} + 4\mathfrak{L}_{\text{AdS},\circ}^{-2}\left( L_\circ^{-2}\ell(\ell+2) + \hat{L}_\circ\hat{\ell}(\hat{\ell}+2) - 8L_{\text{AdS},\circ}^{-2} + 9\Lambda \right)\mathcal{M}
$$
$$
+ \frac{4}{3}\mathfrak{L}_{\text{AdS},\circ}^{-2}\left( 28L_\circ^{-2} + 17\Lambda \right)\mathcal{N} + 12\mathfrak{L}_{\text{AdS},\circ}^{-2}\Lambda\mathcal{P}
$$
$$
+ \left( L_\circ^{-2}\ell(\ell+2) + \hat{L}_\circ\hat{\ell}(\hat{\ell}+2) + \frac{20}{3}\mathfrak{L}_{\text{AdS},\circ}^{-2} \right)\mathcal{U} + \frac{28}{3}\mathfrak{L}_{\text{AdS},\circ}^{-2}\left( 1 + \frac{\hat{L}_\circ^{-2}\hat{\ell}(\hat{\ell}+2)}{L_\circ^{-2}\ell(\ell+2)} \right)\mathcal{V} \,,
$$
$$
\tag{C.60}
$$

$$
\Box_0 \mathcal{V} = -4\mathfrak{L}_\circ^{-2}L_\circ^{-2}\ell(\ell+2)\mathcal{N} + \left( L_\circ^{-2}\ell(\ell+2) + \hat{L}_\circ^{-2}\hat{\ell}(\hat{\ell}+2) \right)\mathcal{V} \,,
\tag{C.61}
$$

$$
\Box_0 \mathcal{W} = -2\hat{\mathfrak{L}}_\circ^{-2}\mathcal{H} - 2\hat{\mathfrak{L}}_\circ^{-2}\left( L_\circ^{-2}\ell(\ell+2) + \hat{L}_\circ^{-2}\hat{\ell}(\hat{\ell}+2) - 8L_{\text{AdS},\circ}^{-2} + 6\Lambda \right)\mathcal{M}
$$
$$
+ \hat{\mathfrak{L}}_\circ^{-2}\left( 4\hat{L}_\circ^{-2}\hat{\ell}(\hat{\ell}+2) + 2L_\circ^{-2}\ell(\ell+2) + \frac{64}{3}L_\circ^{-2} + \frac{44}{3}\Lambda \right)\mathcal{N} - 6\hat{\mathfrak{L}}_\circ^{-2}\hat{L}_\circ^{-2}\hat{\ell}(\hat{\ell}+2)\mathcal{P}
$$
$$
- \frac{28}{3}\hat{\mathfrak{L}}_\circ^{-2}\mathcal{U} - \frac{4}{3}\hat{\mathfrak{L}}_\circ^{-2}\left( 7 + \frac{\hat{L}_\circ^{-2}\hat{\ell}(\hat{\ell}+2)}{L_\circ^{-2}\ell(\ell+2)} \right)\mathcal{V} + \left( L_\circ^{-2}\ell(\ell+2) + \hat{L}_\circ^{-2}\hat{\ell}(\hat{\ell}+2) \right)\mathcal{W} \,,
$$
$$
\tag{C.62}
$$

$$\Box_0 \mathcal{H} = \frac{1}{3}\left(3L_\circ^{-2}\ell(\ell+2) + 3\hat{L}_\circ^{-2}\hat{\ell}(\hat{\ell}+2) + 28L_{\mathrm{AdS},\circ}^{-2} - 8\Lambda\right)\mathcal{H} + \frac{16}{3}\left[-2\mathfrak{L}_{\mathrm{AdS},\circ}^{-2}\hat{L}_\circ^{-2}\hat{\ell}(\hat{\ell}+2)\right.$$

$$+ 16L_{\mathrm{AdS},\circ}^{-4} - 2\Lambda\left(12\mathfrak{L}_{\mathrm{AdS},\circ}^{-2} + L_{\mathrm{AdS},\circ}^{-2}\right) - 2L_\circ^{-2}\left(\mathfrak{L}_{\mathrm{AdS},\circ}^{-2}\ell(\ell+2) - 3\Lambda\right)\bigg]\mathcal{M}$$

$$+ \frac{16}{9}\bigg[12L_\circ^{-4}\left(\ell(\ell+2)+4\right) - 74L_{\mathrm{AdS},\circ}^{-2}L_\circ^{-2}\hat{\ell}(\hat{\ell}+2) + 2\Lambda L_\circ^{-2}\left(3\ell(\ell+2)+47\right)$$

$$+ 12\mathfrak{L}_\circ^{-2}\hat{L}_\circ^{-2}\hat{\ell}(\hat{\ell}+2) - \Lambda\left(76\mathfrak{L}_{\mathrm{AdS},\circ}^{-2} - 33L_{\mathrm{AdS},\circ}^{-2}\right)\bigg]\mathcal{N}$$

$$+ 32\Lambda\left(\mathfrak{L}_\circ^{-2} - \mathfrak{L}_{\mathrm{AdS},\circ}^{-2}\right)\mathcal{P} - \frac{8}{9}\left(3L_\circ^{-2}\ell(\ell+2) + 3\hat{L}_\circ^{-2}\hat{\ell}(\hat{\ell}+2) + 11L_{\mathrm{AdS},\circ}^{-2} - 10\Lambda\right)\mathcal{U}$$

$$+ \frac{8}{9}\bigg[-3L_\circ^{-2}\frac{\ell^2(\ell+4)}{\ell+2} + \left(28\mathfrak{L}_{\mathrm{AdS},\circ}^{-2} + 9L_{\mathrm{AdS},\circ}^{-2} - 3\hat{L}_\circ^{-2}\hat{\ell}(\hat{\ell}+2)\right)\frac{\hat{L}_\circ^{-2}\hat{\ell}(\hat{\ell}+2)}{L_\circ^{-2}\ell(\ell+2)}$$

$$- 12\hat{L}_\circ^{-2}\frac{\hat{\ell}(\hat{\ell}+2)}{\ell+2} + \frac{74\mathfrak{L}_{\mathrm{AdS},\circ}^{-2} - 96\mathfrak{L}_\circ^{-2} + 9\Lambda}{\ell+2}$$

$$- \left(60L_\circ^{-2} + 6\hat{L}_\circ^{-2}\hat{\ell}(\hat{\ell}+2) - 37L_{\mathrm{AdS},\circ}^{-2} + 38\Lambda\right)\frac{\ell}{\ell+2}\bigg]\mathcal{V}\,. \tag{C.63}$$

# D   One-loop corrected spectrum

For small $\ell \leq 1$ or $\hat{\ell} \leq 1$, some coupled scalars are gauged away. The details of why this happens for each such choice of $\ell, \hat{\ell}$ are laid out in [38, Appendix C]. Here, we only list which scalars remain. In section D.1, we provide the small $g_s$ one-loop corrections to the masses of the remaining scalars. In section D.2, we provide the numerical values for the one-loop corrected masses of the coupled scalars for all $g_s$.

## D.1   Small $g_s$

We now list the one-loop corrections to the masses of the spacetime scalars for small $\ell, \hat{\ell}$. Some of the correction terms, however, are too complicated to write out and have been suppressed. The spectrum is symmetric under the exchange of $\ell \leftrightarrow \hat{\ell}$, so we have omitted combinations which are equivalent by symmetry.

$\ell = \hat{\ell} = 0$:

$$\lambda_{4,\circ} L_{\text{AdS},\circ}^2 = \frac{8n_5}{n_5 + \hat{n}_5} + \frac{13n_5 + 11\hat{n}_5}{n_5 + \hat{n}_5} \lambda \frac{L_{\text{AdS}}^2}{\alpha'} g_s^2 + \mathcal{O}(\lambda N g_s^2)^2 \,, \tag{D.1}$$

$$\lambda_{5,\circ} L_{\text{AdS},\circ}^2 = \frac{8\hat{n}_5}{n_5 + \hat{n}_5} + \frac{11n_5 + 13\hat{n}_5}{n_5 + \hat{n}_5} \lambda \frac{L_{\text{AdS}}^2}{\alpha'} g_s^2 + \mathcal{O}(\lambda N g_s^2)^2 \,, \tag{D.2}$$

$$\lambda_{6,\circ} L_{\text{AdS},\circ}^2 = 8 - \lambda \frac{L_{\text{AdS}}^2}{\alpha'} g_s^2 + \mathcal{O}(\lambda N g_s^2)^2 \,. \tag{D.3}$$

$\ell = 1, \hat{\ell} = 0$:

$$\lambda_{4,\circ} L_{\text{AdS},\circ}^2 = \frac{8n_5 + 3\hat{n}_5}{n_5 + \hat{n}_5} + \frac{5(11n_5 + 10\hat{n}_5)}{4(n_5 + \hat{n}_5)} \lambda \frac{L_{\text{AdS}}^2}{\alpha'} g_s^2 + \mathcal{O}(\lambda N g_s^2)^2 \,, \tag{D.4}$$

$$\lambda_{5,\circ} L_{\text{AdS},\circ}^2 = \frac{15\hat{n}_5}{n_5 + \hat{n}_5} + \frac{47n_5 + 62\hat{n}_5}{4(n_5 + \hat{n}_5)} \lambda \frac{L_{\text{AdS}}^2}{\alpha'} g_s^2 + \mathcal{O}(\lambda N g_s^2)^2 \,, \tag{D.5}$$

$$\lambda_{6,\circ} L_{\text{AdS},\circ}^2 = \lambda_6 L_{\text{AdS}}^2 + \mathcal{O}(\lambda N g_s^2) \,, \tag{D.6}$$

$$\lambda_{7,\circ} L_{\text{AdS},\circ}^2 = \lambda_7 L_{\text{AdS}}^2 + \mathcal{O}(\lambda N g_s^2) \,. \tag{D.7}$$

$\ell = 2, \hat{\ell} = 0$:

$$\lambda_{2,\circ} L_{\text{AdS},\circ}^2 = 0 - \frac{1}{3} \lambda \frac{L_{\text{AdS}}^2}{\alpha'} g_s^2 + \mathcal{O}(\lambda N g_s^2)^2 \,, \tag{D.8}$$

$$\lambda_{4,\circ} L_{\text{AdS},\circ}^2 = 8 + 15 \lambda \frac{L_{\text{AdS}}^2}{\alpha'} g_s^2 + \mathcal{O}(\lambda N g_s^2)^2 \,, \tag{D.9}$$

$$\lambda_{5,\circ} L_{\text{AdS},\circ}^2 = \frac{24\hat{n}_5}{n_5 + \hat{n}_5} + \frac{43n_5 + 61\hat{n}_5}{3(n_5 + \hat{n}_5)} \lambda \frac{L_{\text{AdS}}^2}{\alpha'} g_s^2 + \mathcal{O}(\lambda N g_s^2)^2 \,, \tag{D.10}$$

$$\lambda_{6,\circ} L_{\text{AdS},\circ}^2 = \lambda_6 L_{\text{AdS}}^2 + \mathcal{O}(\lambda N g_s^2) \,, \tag{D.11}$$

$$\lambda_{7,\circ} L_{\text{AdS},\circ}^2 = \lambda_6 L_{\text{AdS}}^2 + \mathcal{O}(\lambda N g_s^2) \,. \tag{D.12}$$

$\ell = \hat{\ell} = 1$:

$$\lambda_{3,\circ} L_{\text{AdS},\circ}^2 = 3 + \frac{21}{4} \lambda \frac{L_{\text{AdS}}^2}{\alpha'} g_s^2 + \mathcal{O}(\lambda N g_s^2)^2 \,, \tag{D.13}$$

$$\lambda_{4,\circ} L_{\text{AdS},\circ}^2 = \frac{3(5n_5 + \hat{n}_5)}{n_5 + \hat{n}_5} + \frac{65n_5 + 53\hat{n}_5}{4(n_5 + \hat{n}_5)} \lambda \frac{L_{\text{AdS}}^2}{\alpha'} g_s^2 + \mathcal{O}(\lambda N g_s^2)^2 \,, \tag{D.14}$$

$$\lambda_{5,\circ} L_{\text{AdS},\circ}^2 = \frac{3(n_5 + 5\hat{n}_5)}{n_5 + \hat{n}_5} + \frac{53n_5 + 65\hat{n}_5}{4(n_5 + \hat{n}_5)} \lambda \frac{L_{\text{AdS}}^2}{\alpha'} g_s^2 + \mathcal{O}(\lambda N g_s^2)^2 \,, \tag{D.15}$$

$$\lambda_{6,\circ} L_{\text{AdS},\circ}^2 = 15 + \frac{5}{2} \lambda \frac{L_{\text{AdS}}^2}{\alpha'} g_s^2 + \mathcal{O}(\lambda N g_s^2)^2 \,, \tag{D.16}$$

$$\lambda_{7,\circ} L_{\text{AdS},\circ}^2 = -1 + 2 \lambda \frac{L_{\text{AdS}}^2}{\alpha'} g_s^2 + \mathcal{O}(\lambda N g_s^2)^2 \,. \tag{D.17}$$

$\ell = 2, \hat{\ell} = 1$:

$$\lambda_{2,\circ} L^2_{\text{AdS},\circ} = \frac{3n_5}{n_5 + \hat{n}_5} + \frac{14n_5 + 5\hat{n}_5}{12(n_5 + \hat{n}_5)} \lambda \frac{L^2_{\text{AdS}}}{\alpha'} g_s^2 + \mathcal{O}(\lambda N g_s^2)^2 \,, \tag{D.18}$$

$$\lambda_{3,\circ} L^2_{\text{AdS},\circ} = \frac{3n_5 + 8\hat{n}_5}{n_5 + \hat{n}_5} + \frac{26n_5 + 31\hat{n}_5}{4(n_5 + \hat{n}_5)} \lambda \frac{L^2_{\text{AdS}}}{\alpha'} g_s^2 + \mathcal{O}(\lambda N g_s^2)^2 \,, \tag{D.19}$$

$$\lambda_{4,\circ} L^2_{\text{AdS},\circ} = \frac{15n_5 + 8\hat{n}_5}{n_5 + \hat{n}_5} + \frac{7(10n_5 + 9\hat{n}_5)}{4(n_5 + \hat{n}_5)} \lambda \frac{L^2_{\text{AdS}}}{\alpha'} g_s^2 + \mathcal{O}(\lambda N g_s^2)^2 \,, \tag{D.20}$$

$$\lambda_{5,\circ} L^2_{\text{AdS},\circ} = \frac{3(n_5 + 8\hat{n}_5)}{n_5 + \hat{n}_5} + \frac{190n_5 + 253\hat{n}_5}{12(n_5 + \hat{n}_5)} \lambda \frac{L^2_{\text{AdS}}}{\alpha'} g_s^2 + \mathcal{O}(\lambda N g_s^2)^2 \,, \tag{D.21}$$

$$\lambda_{6,\circ} L^2_{\text{AdS},\circ} = \lambda_6 L^2_{\text{AdS}} + \mathcal{O}(\lambda N g_s^2) \,, \tag{D.22}$$

$$\lambda_{7,\circ} L^2_{\text{AdS},\circ} = \lambda_7 L^2_{\text{AdS}} + \mathcal{O}(\lambda N g_s^2) \,. \tag{D.23}$$

$\ell = \hat{\ell} = 2$:

$$\lambda_{1,\circ} L^2_{\text{AdS},\circ} = \frac{8\hat{n}_5}{n_5 + \hat{n}_5} + \frac{5n_5 + 11\hat{n}_5}{3(n_5 + \hat{n}_5)} \lambda \frac{L^2_{\text{AdS}}}{\alpha'} g_s^2 + \mathcal{O}(\lambda N g_s^2)^2 \,, \tag{D.24}$$

$$\lambda_{2,\circ} L^2_{\text{AdS},\circ} = \frac{8n_5}{n_5 + \hat{n}_5} + \frac{11n_5 + 5\hat{n}_5}{3(n_5 + \hat{n}_5)} \lambda \frac{L^2_{\text{AdS}}}{\alpha'} g_s^2 + \mathcal{O}(\lambda N g_s^2)^2 \,, \tag{D.25}$$

$$\lambda_{3,\circ} L^2_{\text{AdS},\circ} = 8 + 9\lambda \frac{L^2_{\text{AdS}}}{\alpha'} g_s^2 + \mathcal{O}(\lambda N g_s^2)^2 \,, \tag{D.26}$$

$$\lambda_{4,\circ} L^2_{\text{AdS},\circ} = \frac{8(3n_5 + \hat{n}_5)}{n_5 + \hat{n}_5} + \frac{67n_5 + 55\hat{n}_5}{3(n_5 + \hat{n}_5)} \lambda \frac{L^2_{\text{AdS}}}{\alpha'} g_s^2 + \mathcal{O}(\lambda N g_s^2)^2 \,, \tag{D.27}$$

$$\lambda_{5,\circ} L^2_{\text{AdS},\circ} = \frac{8(n_5 + 3\hat{n}_5)}{n_5 + \hat{n}_5} + \frac{55n_5 + 67\hat{n}_5}{3(n_5 + \hat{n}_5)} \lambda \frac{L^2_{\text{AdS}}}{\alpha'} g_s^2 + \mathcal{O}(\lambda N g_s^2)^2 \,, \tag{D.28}$$

$$\lambda_{6,\circ} L^2_{\text{AdS},\circ} = 24 + 7\lambda \frac{L^2_{\text{AdS}}}{\alpha'} g_s^2 + \mathcal{O}(\lambda N g_s^2)^2 \,, \tag{D.29}$$

$$\lambda_{7,\circ} L^2_{\text{AdS},\circ} = 0 + 5\lambda \frac{L^2_{\text{AdS}}}{\alpha'} g_s^2 + \mathcal{O}(\lambda N g_s^2)^2 \,. \tag{D.30}$$

## D.2  Large $g_s$

All the scalar masses are sigmoid functions with respect to $g_s^2$. This means that if the first order correction is positive, the scalar will always mass up in the large $g_s^2$ limit and vice versa. We are interested in whether these scalars fall below the BF bound. Therefore, we are only concerned with the scalars that have a negative first order correction in $g_s^2$ which consequently risk falling below the BF bound. We provide the numerical analysis for the masses of these scalars in the large $g_s^2$ limit in the plots below and show that none of them cross the BF bound.

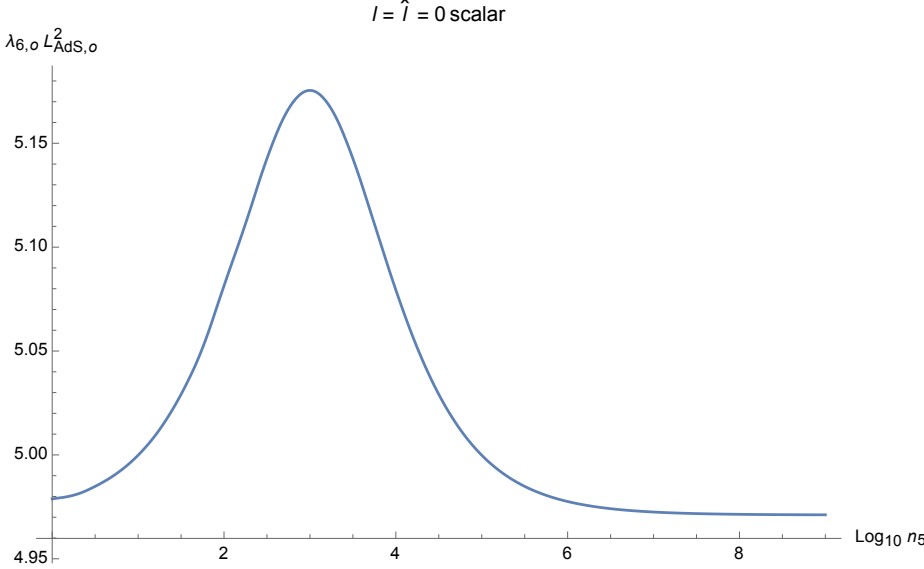

Figure 6: We fix $\hat{n}_5 = 10^3$ and plot $\ell = 0, \hat{\ell} = 0$ scalar mass $\lambda_{6,\circ}L^2_{\text{AdS},\circ}$ versus $n_5$ in the large $g_s^2$ (small $n_1$) limit.

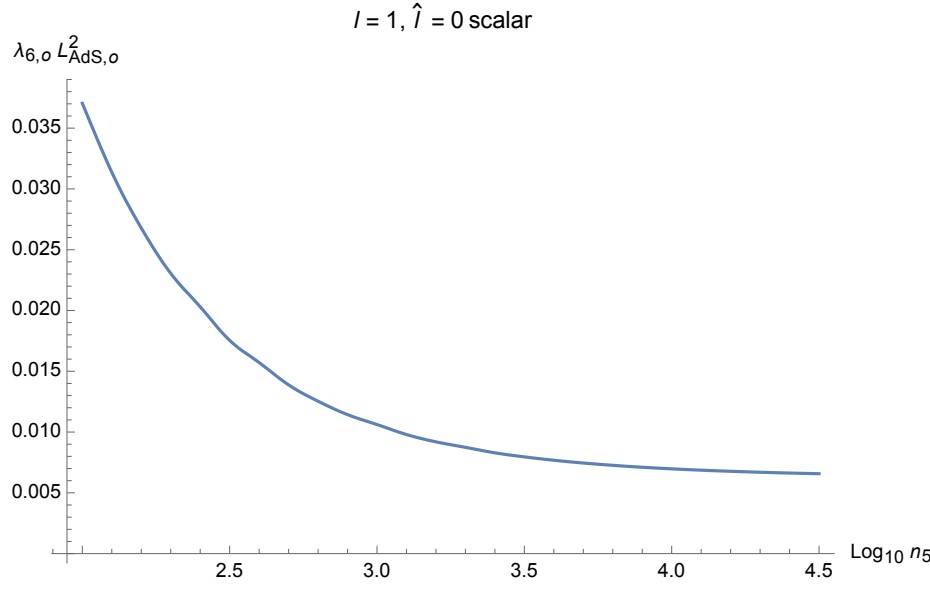

Figure 7: We fix $\hat{n}_5 = 10^3$ and plot $\ell = 1, \hat{\ell} = 0$ scalar mass $\lambda_{6,\circ}L^2_{\text{AdS},\circ}$ versus $n_5$ in the large $g_s^2$ (small $n_1$) limit.

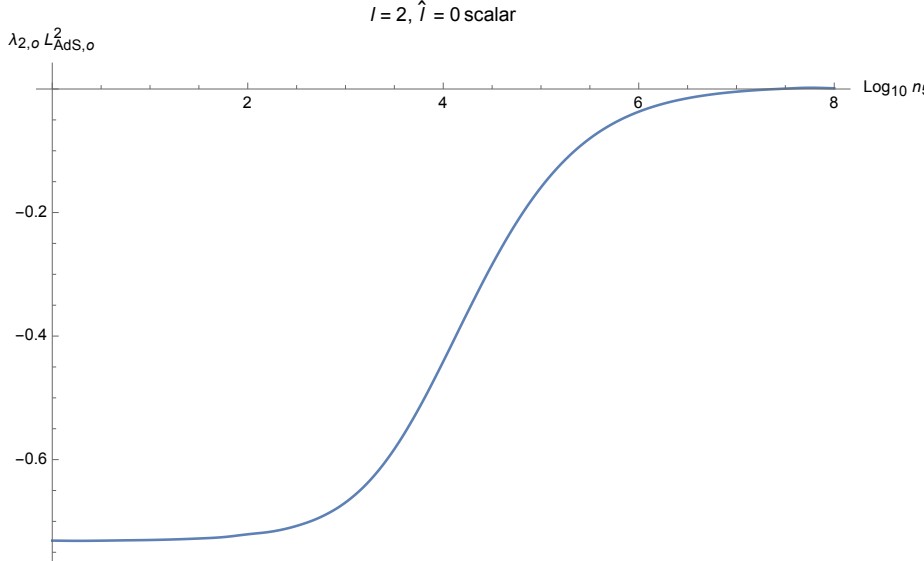

Figure 8: We fix $\hat{n}_5 = 10^3$ and plot $\ell = 2, \hat{\ell} = 0$ scalar mass $\lambda_{2,\circ}L^2_{\text{AdS},\circ}$ versus $n_5$ in the large $g_s^2$ (small $n_1$) limit.

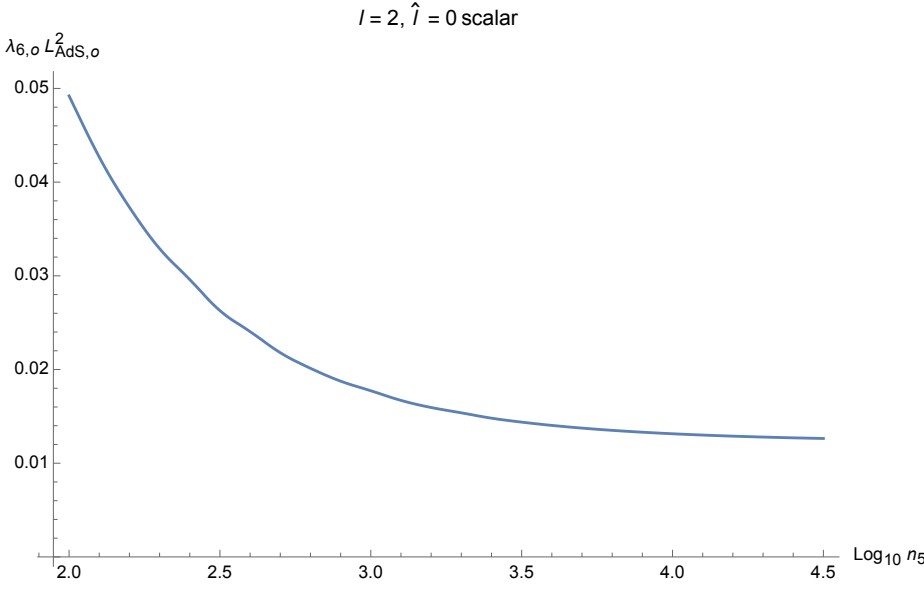

Figure 9: We fix $\hat{n}_5 = 10^3$ and plot $\ell = 2, \hat{\ell} = 0$ scalar mass $\lambda_{6,\circ}L^2_{\text{AdS},\circ}$ versus $n_5$ in the large $g_s^2$ (small $n_1$) limit.

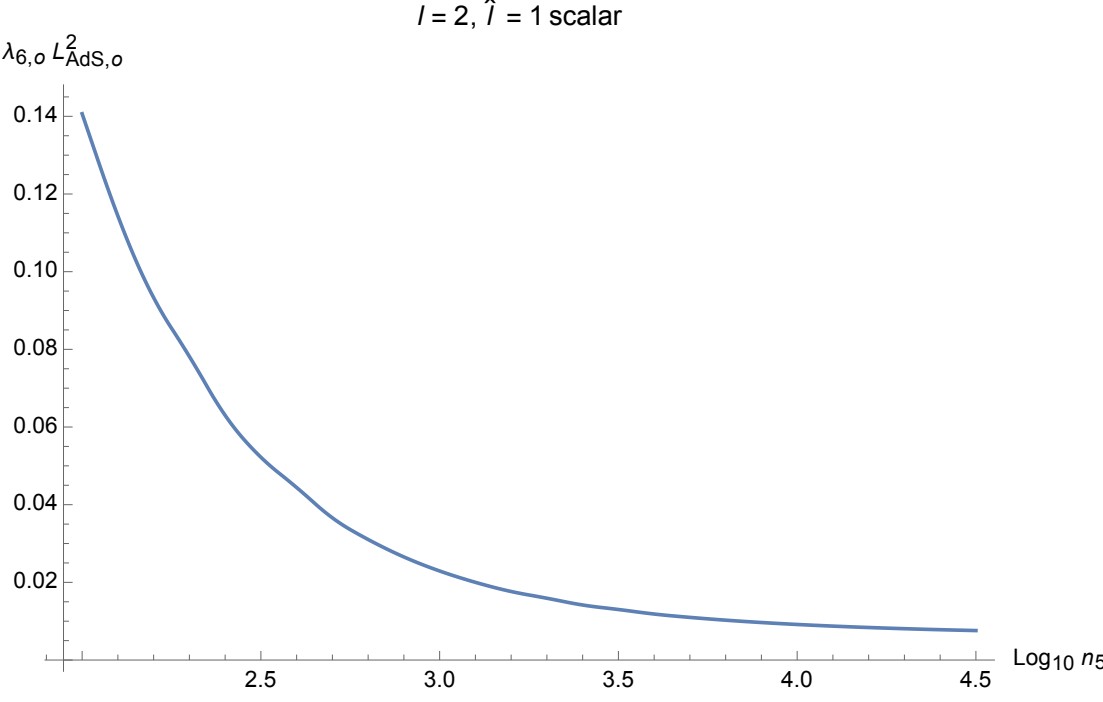

Figure 10: We fix $\hat{n}_5 = 10^3$ and plot $\ell = 2, \hat{\ell} = 1$ scalar mass $\lambda_{6,\circ}L^2_{\text{AdS},\circ}$ versus $n_5$ in the large $g_s^2$ (small $n_1$) limit.

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
