# Peer review of "Non-Supersymmetric AdS from String Theory"

_SciPost Physics_

## Round 2 · Referee Report · Anonymous (Referee 1) · 2023-5-5

Report

In this paper, the authors study non-supersymmetric anti-de Sitter (AdS) flux compactifications of the O(16)xO(16) heterotic string. This is the unique ten-dimensional heterotic theory without perturbative tachyons, which makes it an ideal candidate to explore non-supersymmetric vacua - even more so since the worldsheet description of heterotic theories is usually under better control with respect to orientifold models.

Combining spacetime effective field theory techniques with worldsheet considerations, the authors perform a very clear and detailed analysis of various cases. A particularly important distinction is that between vacua that exist in classical string theory, whose worldsheet description is also under control, and vacua that are "intrinsically quantum", in the sense that string loop effects have to be included in order to obtain them. In addition, the authors show that de Sitter vacua cannot be obtained by including such effects.

After presenting the vacuum solutions, the authors delve into a detailed analysis of perturbative stability, showing the absence of tachyons below the Breitenlohner-Freedman bound. Some perturbations are argued to be stable indirectly, based on gauge invariance and worldsheet considerations, but the quantitative analysis is nonetheless complete and the exposition contains all the necessary details.

Overall I recommend the paper for publication. A few points that could be improved are the following:

  1. The discussion on the Casimir energy in the Summary could be slightly fleshed out (although the author do mention that it merits a separate discussion in future work), expanding on why it is relevant and roughly what could happen including this contribution in their analysis.

  2. The reader could benefit from a concluding section after the heavily technical analysis of the final sections. Either that, or the outlook and conclusions contained in the Summary could be expanded to cover how these results fit in the state of the field on a more conceptual level.

---

## Round 2 · Referee Report · Anonymous (Referee 2) · 2023-6-25

Strengths

1 - This is an interesting paper, exploring the stability of non-supersymmetric AdS vacua in string theory, which is a long-standing, ongoing and important debate.
2 - While a lot of work has previously focused on low-energy supersymmetry breaking of the type II strings or M-theory, this paper instead focuses on the O(16)xO(16) non-supersymmetric string theory, where much less is known.
3 - The paper studies the interesting setup of "intrinsically quantum" vacua, where effective potentials from the string tree-level and one-loop balance against one another to generate the vacuum. Such setups have hardly been investigated in the past and this paper leads an interesting discussion on this topic.
4 - There are several interesting effects that appear in the vacua studied. For example, the vacua seem to not admit a strong-coupling limit by adjusting the flux parameters. Thus, this paper really opens up a new way of creating potentially interesting weakly-coupled string vacua.
5 - The paper explores a new way of constructing string vacua (string-scale SUSY breaking, inherently quantum, ...) and does a reasonable job at addressing the obvious questions, such as perturbative stability. There are many other open questions, such as non-perturbative instabilities, but it is unreasonable to expect this paper to address all these issues.

Weaknesses

The paper has just a few minor weaknesses.
1 - The discussion of the 3-dimensional effective action in section 2 is somewhat confusing. Firstly, all well-controlled AdS vacua in string theory do not admit scale separation, and thus no lower-dimensional local effective action. This is certainly the case for the supersymmetric AdS_3 x S^3 x S^3 x S^1 vacua of heterotic and type II string theories. Why do the authors refer to an effective action here? Because there are no effective actions, the lower-dimensional actions must be constructed as consistent truncations, which is difficult and technical, especially if one wants the supersymmetry of the AdS vacuum to be reflected in the lower-dimensional supergravity. However, the authors do not seem to require such a rich 3-dimensional action. In fact, they propose in (2.4) an Ansatz for the metric, which might serve as a consistent truncation. The Ansatz (2.4) indeed looks like it might be keeping all SO(4) x SO(4) singlets, which would likely guarantee consistency. However, I wonder why there wouldn't then also be a warp factor in front of the 3-d metric in (2.4). Or is this implied by the notation? If so, it would be good to spell this out precisely. Indeed, it would be good to know if the solution (2.17) and the potential (2.20), (2.21) (and (4.1)) satisfy the 10-d equations of motion. Since the authors study the 10-d equations of motion in section 5, they can probably check this easily. It would be good for them to clarify whether the 3-d action they are studying is such a consistent truncation.
2 - It is unclear to me why the authors say that the perturbative stability of [10] is still being assessed. [10] proves the perturbative stability of those vacua. Perhaps the author refer to potential marginal multi-trace operators that could have non-vanishing beta functions in the vacua of [10]. If so, it would be good to clear that up.
3 - The authors may also want to include skew-whiffed vacua, e.g. reviewed in Duff, Nilsson, Pope, Phys.Rept. 130 (1986) 1-142, which are perturbatively stable non-SUSY vacua in supergravity.

Report

This is a very interesting paper and should be accepted for publication.

Requested changes

See weaknesses above.

---

## Editorial Decision

resubmitted